# For SALE: State-Action Representation Learning for Deep Reinforcement Learning

**Scott Fujimoto**
Mila, McGill University

**Wei-Di Chang**
McGill University

**Edward J. Smith**
McGill University

**Shixiang Shane Gu**
Google DeepMind

**Doina Precup**
Mila, McGill University

**David Meger**
Mila, McGill University

## Abstract

In the field of reinforcement learning (RL), representation learning is a proven tool for complex image-based tasks, but is often overlooked for environments with low-level states, such as physical control problems. This paper introduces SALE, a novel approach for learning embeddings that model the nuanced interaction between state and action, enabling effective representation learning from low-level states. We extensively study the design space of these embeddings and highlight important design considerations. We integrate SALE and an adaptation of checkpoints for RL into TD3 to form the TD7 algorithm, which significantly outperforms existing continuous control algorithms. On OpenAI gym benchmark tasks, TD7 has an average performance gain of 276.7% and 50.7% over TD3 at 300k and 5M time steps, respectively, and works in both the online and offline settings.

## 1 Introduction

Reinforcement learning (RL) is notoriously sample inefficient, particularly when compared to more straightforward paradigms in machine learning, such as supervised learning. One possible explanation is the usage of the Bellman equation in most off-policy RL algorithms [Mnih et al., 2015, Lillicrap et al., 2015], which provides a weak learning signal due to an approximate and non-stationary learning target [Fujimoto et al., 2022].

A near-universal solution to sample inefficiency in deep learning is representation learning, whereby intermediate features are learned to capture the underlying structure and patterns of the data. These features can be found independently from the downstream task and considerations such as the learning horizon and dynamic programming. While feature learning of this type has found some success in the RL setting, it has been mainly limited to vision-based environments [Jaderberg et al., 2017, Oord et al., 2018, Anand et al., 2019, Laskin et al., 2020, Stooke et al., 2021, Yarats et al., 2022].

On the other hand, the application of representation learning to low-level states is much less common. At first glance, it may seem unnecessary to learn a representation over an already-compact state vector. However, we argue that the difficulty of a task is often defined by the complexity of the underlying dynamical system, rather than the size of the observation space. This means that regardless of the original observation space, there exists an opportunity to learn meaningful features by capturing the interaction between state and action.

**SALE.** In this paper, we devise state-action learned embeddings (SALE), a method that learns embeddings jointly over both state and action by modeling the dynamics of the environment in latent space. Extending prior work [Ota et al., 2020], we introduce three important design considerations

---

Corresponding author: scott.fujimoto@mail.mcgill.ca

37th Conference on Neural Information Processing Systems (NeurIPS 2023).

when learning a state-action representation online. Most importantly, we observe the surprising effect of extrapolation error [Fujimoto et al., 2019] when significantly expanding the action-dependent input and introduce a simple clipping technique to mitigate it.

**Design study.** Learning to model environment dynamics in latent space is a common approach for feature learning which has been widely considered [Watter et al., 2015, Ha and Schmidhuber, 2018, Hafner et al., 2019, Gelada et al., 2019, Schwarzer et al., 2020], with many possible variations in design. Consequently, the optimal design decision is often unclear without considering empirical performance. To this end, we perform an extensive empirical evaluation over the design space, with the aim of discovering which choices are the most significant contributors to final performance.

**Checkpoints.** Next, we explore the usage of checkpoints in RL. Similar to representation learning, early stopping and checkpoints are standard techniques used to enhance the performance of deep learning models. A similar effect can be achieved in RL by fixing each policy for multiple training episodes, and then at test time, using the highest-performing policy observed during training.

**TD7.** We combine TD3 with our state-action representation learning method SALE, the aforementioned checkpoints, prioritized experience replay [Fujimoto et al., 2020], and a behavior cloning term (used only for offline RL) [Fujimoto and Gu, 2021] to form the TD7 (TD3+4 additions) algorithm. We benchmark the TD7 algorithm in both the online and offline RL setting. TD7 significantly outperforms existing methods without the additional complexity from competing methods such as large ensembles, additional updates per time step, or per-environment hyperparameters. Our key improvement, SALE, works in tandem with most RL methods and can be used to enhance existing approaches in both the online and offline setting. Our code is open-sourced[1].

## 2 Related Work

**Representation learning.** Representation learning has several related interpretations in RL. Historically, representation learning referred to *abstraction*, mapping an MDP to a smaller one via bisimulation or other means [Li et al., 2006, Ferns et al., 2011, Zhang et al., 2020]. For higher-dimensional spaces, the notion of true abstraction has been replaced with *compression*, where the intent is to embed the observation space (such as images) into a smaller manageable latent vector [Watter et al., 2015, Finn et al., 2016, Gelada et al., 2019]. Representation learning can also refer to *feature learning*, where the objective is to learn features that capture relevant aspects of the environment or task, via auxiliary rewards or alternate training signals [Sutton et al., 2011, Jaderberg et al., 2017, Riedmiller et al., 2018, Lin et al., 2019]. In recent years, representation learning in RL often refers to both compression and feature learning, and is commonly employed in image-based tasks [Kostrikov et al., 2020, Yarats et al., 2021, Liu et al., 2021, Cetin et al., 2022] where the observation space is characterized by its high dimensionality and the presence of redundant information.

Representation learning by predicting future states draws inspiration from a rich history [Dayan, 1993, Littman and Sutton, 2001], spanning many approaches in both model-free RL [Munk et al., 2016, Van Hoof et al., 2016, Zhang et al., 2018, Gelada et al., 2019, Schwarzer et al., 2020, Fujimoto et al., 2021, Ota et al., 2020, 2021] and model-based RL in latent space [Watter et al., 2015, Finn et al., 2016, Karl et al., 2017, Ha and Schmidhuber, 2018, Hansen et al., 2022, Hafner et al., 2019, 2023]. Another related approach is representation learning over actions [Tennenholtz and Mannor, 2019, Chandak et al., 2019, Whitney et al., 2020]. Our key distinction from many previous approaches is the emphasis on learning joint representations of both state and action.

Methods which do learn state-action representations, by auxiliary rewards to the value function [Liu et al., 2021], or MDP homomorphisms [Ravindran, 2004, van der Pol et al., 2020a,b, Rezaei-Shoshtari et al., 2022] emphasize abstraction more than feature learning. Our approach can be viewed as an extension of OFENet [Ota et al., 2020], which also learns a state-action embedding. We build off of OFENet and other representation learning methods by highlighting crucial design considerations and addressing the difficulties that arise when using decoupled state-action embeddings. Our resulting improvements are reflected by significant performance gains in benchmark tasks.

**Stability in RL.** Stabilizing deep RL algorithms has been a longstanding challenge, indicated by numerous empirical studies that highlight practical concerns associated with deep RL methods [Henderson et al., 2017, Engstrom et al., 2019]. Our use of checkpoints is most closely related to stabilizing

---

[1] https://github.com/sfujim/TD7

policy performance via safe policy improvement [Schulman et al., 2015, 2017, Laroche et al., 2019], as well as the combination of evolutionary algorithms [Salimans et al., 2017, Mania et al., 2018] with RL [Khadka and Tumer, 2018, Pourchot and Sigaud, 2018], where the checkpoint resembles the fittest individual and the mutation is defined exclusively by the underlying RL algorithm.

# 3 Background

In Reinforcement learning (RL) problems are framed as a Markov decision process (MDP). An MDP is a 5-tuple $(S, A, R, p, \gamma)$ with state space $S$, action space $A$, reward function $R$, dynamics model $p$, and discount factor $\gamma$, where the objective is to find a policy $\pi : S \to A$, a mapping from state $s \in S$ to action $a \in A$, which maximizes the return $\sum_{t=1}^{\infty} \gamma^{t-1} r_t$, the discounted sum of rewards $r$ obtained when following the policy. RL algorithms commonly use a value function $Q^{\pi}(s, a) := \mathbb{E}\left[\sum_{t=1}^{\infty} \gamma^{t-1} r_t | s_0 = s, a_0 = a\right]$, which models the expected return, starting from an initial state $s$ and action $a$.

# 4 State-Action Representation Learning

In this section, we introduce state-action learned embeddings (SALE) (Figure 1). We begin with the basic outline of SALE and then discuss three important considerations in how SALE is implemented. We then perform an extensive empirical evaluation on the design space to highlight the critical choices when learning embeddings from the dynamics of the environment.

## 4.1 State-Action Learned Embeddings

The objective of SALE is to discover learned embeddings $(z^{sa}, z^s)$ which capture relevant structure in the observation space, as well as the transition dynamics of the environment. To do so, SALE utilizes a pair of encoders $(f, g)$ where $f(s)$ encodes the state $s$ into the state embedding $z^s$ and $g(z^s, a)$ jointly encodes both state $s$ and action $a$ into the state-action embedding $z^{sa}$:

$$z^s := f(s), \qquad z^{sa} := g(z^s, a). \tag{1}$$

The embeddings are split into state and state-action components so that the encoders can be trained with a dynamics prediction loss that solely relies on the next state $s'$, independent of the next action or current policy. As a result, the encoders are jointly trained using the mean squared error (MSE) between the state-action embedding $z^{sa}$ and the embedding of the next state $z^{s'}$:

$$\mathcal{L}(f, g) := \left(g(f(s), a) - |f(s')|_{\times}\right)^2 = \left(z^{sa} - |z^{s'}|_{\times}\right)^2, \tag{2}$$

where $|\cdot|_{\times}$ denotes the stop-gradient operation. The embeddings are designed to model the underlying structure of the environment. However, they may not encompass all relevant information needed by the value function and policy, such as features related to the reward, current policy, or task horizon. Accordingly, we concatenate the embeddings with the original state and action, allowing the value and policy networks to learn relevant internal representations for their respective tasks:

$$Q(s, a) \to Q(z^{sa}, z^s, s, a), \qquad \pi(s) \to \pi(z^s, s). \tag{3}$$

The encoders $(f, g)$ are trained online and concurrently with the RL agent (updated at the same frequency as the value function and policy), but are decoupled (gradients from the value function and policy are not propagated to $(f, g)$). Although the embeddings are learned by considering the dynamics of the environment, their purpose is solely to improve the input to the value function and policy, and not to serve as a world model for planning or estimating rollouts.

There are three additional considerations in how SALE is implemented in practice.

**Normalized embeddings.** The minimization of distances in embedding space can result in instability due to either monotonic growth or collapse to a redundant representation [Gelada et al., 2019]. To combat this risk, we introduce AvgL1Norm, a normalization layer that divides the input vector by its average absolute value in each dimension, thus keeping the relative scale of the embedding constant throughout learning. Let $x_i$ be the $i$-th dimension of an $N$-dimensional vector $x$, then

$$\text{AvgL1Norm}(x) := \frac{x}{\frac{1}{N} \sum_i |x_i|}. \tag{4}$$

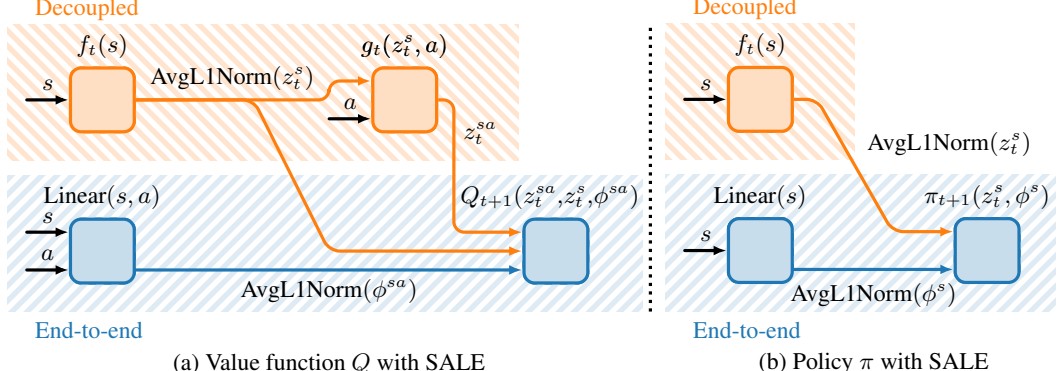

Figure 1: **Diagram of State-Action Learned Embeddings (SALE).** SALE uses encoders $(f, g)$ to output embeddings $(z^s, z^{sa})$ to enhance the input of the value function $Q$ and policy $\pi$. $\phi$ denotes the output of the corresponding linear layer. ■ The encoders $(f, g)$ are jointly trained to predict the next state embedding (where $| \cdot |_\times$ denotes the stop-gradient operation), decoupled from the training of the value function and policy (Equation 2). ■ The end-to-end linear layers are trained with gradients from the corresponding network. AvgL1Norm is used to keep the scale of each of the inputs to the value function and policy constant.

AvgL1Norm is applied to the state embedding $z^s$. Similar to the normalized loss functions used by SPR [Schwarzer et al., 2020] and BYOL [Grill et al., 2020], AvgL1Norm protects from monotonic growth, but also keeps the scale of the downstream input constant without relying on updating statistics (e.g. BatchNorm [Ioffe and Szegedy, 2015]). This is important for our approach as the embeddings are trained independently from the value function and policy. AvgL1Norm is not applied to the state-action embedding $z^{sa}$, as it is trained to match the normalized next state embedding $z^{s'}$.

We also apply AvgL1Norm to the state and action inputs (following a linear layer) to the value function $Q$ and policy $\pi$, to keep them at a similar scale to the learned embeddings. The input to the value function and policy then becomes:

$$Q(z^{sa}, z^s, \text{AvgL1Norm}(\text{Linear}(s, a))), \qquad \pi(z^s, \text{AvgL1Norm}(\text{Linear}(s))). \qquad (5)$$

Unlike the embeddings $(z^s, z^{sa})$, these linear layers are learned end-to-end, and can consequently be viewed as an addition to the architecture of the value function or policy.

**Fixed embeddings.** Since an inconsistent input can cause instability, we freeze the embeddings used to train the current value and policy networks. This means at the iteration $t + 1$, the input to the current networks $(Q_{t+1}, \pi_{t+1})$ uses embeddings $(z_t^{sa}, z_t^s)$ from the encoders $(f_t, g_t)$ at the previous iteration $t$. The value function and policy are thus updated by:

$$Q_{t+1}(z_t^{sa}, z_t^s, s, a) \approx r + \gamma Q_t(z_{t-1}^{s'a'}, z_{t-1}^{s'}, s', a'), \qquad \text{where } a' \sim \pi_t(z_{t-1}^{s'}, s'), \qquad (6)$$

$$\pi_{t+1}(z_t^s, s) \approx \underset{\pi}{\text{argmax}}\, Q_{t+1}(z_t^{sa}, z_t^s, s, a), \qquad \text{where } a \sim \pi(z_t^s, s). \qquad (7)$$

The current value function $Q_{t+1}$ is also trained with respect to the previous value function $Q_t$, known as a target network [Mnih et al., 2015]. The current embeddings $z_{t+1}^s$ and $z_{t+1}^{sa}$ are trained with Equation 2, using a target $z_{t+1}^{s'}$ (hence, without a target network). Every $n$ steps the iteration is incremented and all target networks are updated simultaneously:

$$Q_t \leftarrow Q_{t+1}, \qquad \pi_t \leftarrow \pi_{t+1}, \qquad (f_{t-1}, g_{t-1}) \leftarrow (f_t, g_t), \qquad (f_t, g_t) \leftarrow (f_{t+1}, g_{t+1}). \qquad (8)$$

**Clipped Values.** Extrapolation error is the tendency for deep value functions to extrapolate to unrealistic values on state-actions pairs which are rarely seen in the dataset [Fujimoto et al., 2019]. Extrapolation error has a significant impact in offline RL, where the RL agent learns from a given dataset rather than collecting its own experience, as the lack of feedback on overestimated values can result in divergence.

Surprisingly, we observe a similar phenomenon in online RL, when increasing the number of dimensions in the state-action input to the value function, as illustrated in Figure 2. Our hypothesis is that the state-action embedding $z^{sa}$ expands the action input and makes the value function more

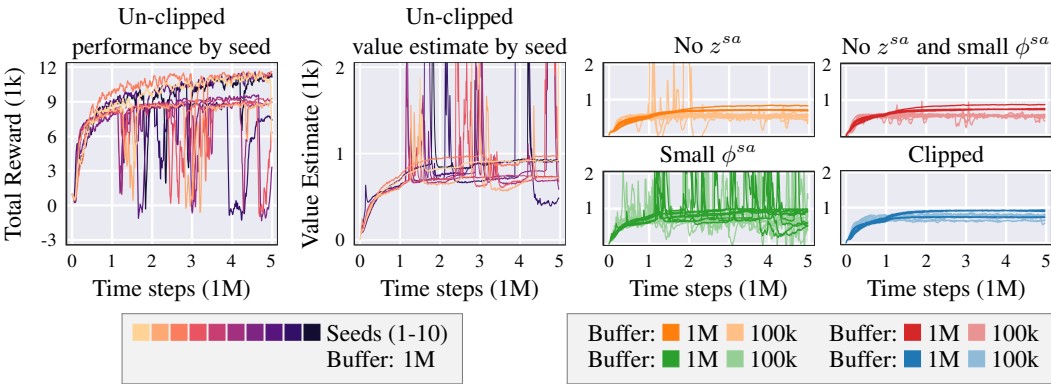

Figure 2: **Extrapolation error can occur in online RL when using state-action representation learning.** All figures use the Ant environment. $\phi^{sa}$ corresponds to the output of the linear layer (Linear$(s,a) = \phi^{sa}$) (Equation 5). Both embeddings and $\phi^{sa}$ have a default dimension size of 256. Small $\phi^{sa}$ means that Dim$(\phi^{sa})$ is set to 16. No $z^{sa}$ means the value function input is $Q(z^s, s, a)$. ▮▮▮ The default performance and value estimate of 10 individual seeds without value clipping. While the performance trends upwards there are large dips in reward, which correspond with jumps in the estimated value. ▮▮/▮▮/▮▮ Varying the input dimension can improve or harm stability of the value estimate. The severity is impacted by the replay buffer size (1M or 100k). The state embedding $z^s$ is left unchanged in all settings, showing that the state-action embedding $z^{sa}$ and the linear layer over the state-action input $\phi^{sa}$ are the primary contributors to the extrapolation error. This shows the potential negative impact from increasing the dimension size of an input which relies on a potentially unseen action. ▮▮ Clipping stabilizes the value estimate, without modifying the input dimension size (Equation 9).

likely to over-extrapolate on unknown actions. We show in Figure 2 that the dimension size of $z^{sa}$ as well as the state-action input plays an important role in the stability of value estimates.

Fortunately, extrapolation error can be combated in a straightforward manner in online RL, where poor estimates are corrected by feedback from interacting with the environment. Consequently, we only need to stabilize the value estimate until the correction occurs. This can be achieved in SALE by tracking the range of values in the dataset $D$ (estimated over sampled mini-batches during training), and then bounding the target used in Equation 6 by the range:

$$Q_{t+1}(s,a) \approx r + \gamma \, \text{clip}\left(Q_t(s', a'), \min_{(s,a) \in D} Q_t(s,a), \max_{(s,a) \in D} Q_t(s,a)\right). \tag{9}$$

Additional discussion of extrapolation error, experimental details, and ablation of the proposed value clipping in SALE can be found in Appendix E & D.

## 4.2 Evaluating Design Choices

The effectiveness of learning embeddings by modeling the dynamics of the environment is a natural consequence of the relationship between the value function and future states. However, there are many design considerations for which all alternatives are potentially valid and the approach adopted differs among related methods in the literature. In this section, we perform an extensive study over the design space to (1) show SALE uses the correct and highest performing set of choices, and (2) better understand which choices are the biggest contributors to performance when using SALE.

In Figure 3 we display the mean percent loss when modifying SALE in the TD7 algorithm (to be fully introduced in Section 5.2). The percent loss is determined from the average performance at 1M time steps, over 10 seeds and five benchmark environments (HalfCheetah, Hopper, Walker2d, Ant, Humanoid) [Brockman et al., 2016]. A more detailed description of each variation and complete learning curves can be found in Appendix D.

**Learning target.** TD7 trains the encoders by minimizing the MSE between the state-action embedding $z^{sa}$ and a learning target of the next state embedding $z^{s'}$ (Equation 2). We test several alternate learning targets. OFENet uses the next state $s'$ as the target [Ota et al., 2020] while SPR [Schwarzer et al., 2020] uses the embedding $z^{s'}_{\text{target}}$ from a target network obtained with an exponential moving average with weight 0.01. Drawing inspiration from Bisimulation metrics [Ferns et al., 2011], Deep-MDP [Gelada et al., 2019] use an objective that considers both the next state embedding $z^{s'}$ and the

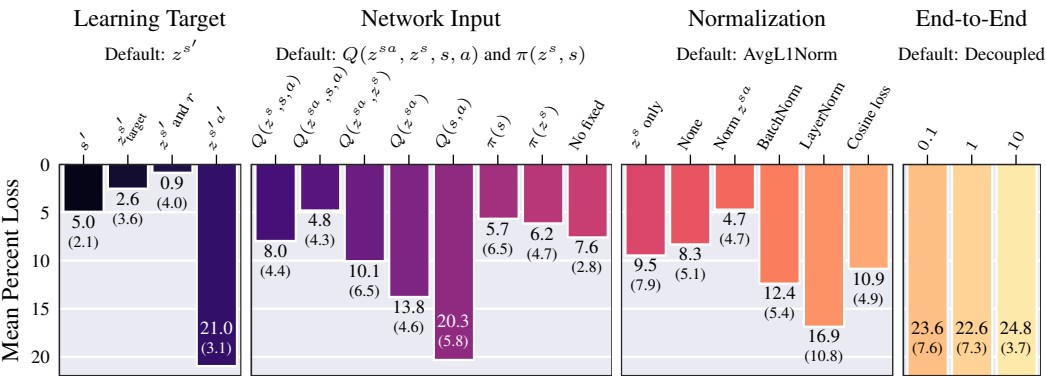

Figure 3: The mean percent loss from using alternate design choices in TD7 at 1M time steps, over 10 seeds and the five benchmark MuJoCo environments. Bracketed values describe the range of the 95% confidence interval around the mean. Percent loss is computed against TD7 where the default choices correspond to a percent loss of 0. See Section 4.2 for a description of each design choice and key observations. See the Appendix for further implementation-level details.

reward $r$. We test including a prediction loss on the reward by having the encoder $g$ output both $z^{sa}$ and $r^{\text{pred}}$ where $r^{\text{pred}}$ is trained with the MSE to the reward $r$. Finally, we test the next state-action embedding $z^{s'a'}$ as the target, where the action $a'$ is sampled from the target policy.

> $\Rightarrow$ All learning targets based on the next state $s'$ perform similarly, although using the embedding $z^{s'}$ further improves the performance. On the contrary, the next state-action embedding $z^{s'a'}$ performs much worse as a target, highlighting that signal based on the non-stationary policy can harm learning. Including the reward as a signal has little impact on performance.

**Network input.** In our approach, the learned embeddings $(z^{sa}, z^s)$ are appended to the state and action input to the value function $Q(z^{sa}, z^s, s, a)$ and policy $\pi(z^s, s)$ (Equation 3). We attempt different combinations of input to both networks. We also evaluate replacing the fixed embeddings (Equations 6 & 7), with the non-static current embeddings $(z^{sa}_{t+1}, z^s_{t+1})$.

> $\Rightarrow$ The added features have a greater impact on the value function than the policy, but are beneficial for both networks. All components of the value function input $(z^{sa}, z^s, s, a)$, are necessary to achieve the highest performance. While the state-action embedding $z^{sa}$ is a useful representation for value learning, it is only trained to predict the next state and may overlook other relevant aspects of the original state-action input $(s, a)$. Solely using the state-action embedding $z^{sa}$ as input leads to poor results, but combining it with the original input $(s, a)$ significantly improves performance.

**Normalization.** TD7 uses AvgL1Norm (Equation 4) to normalize the scale of the state embedding $z^s$, as well as on the state-action input $(s, a)$, following a linear layer (Equation 5). We attempt removing AvgL1Norm on $(s, a)$, removing it entirely, and adding it to the state-action embedding $z^{sa}$. We additionally test swapping AvgL1Norm for BatchNorm [Ioffe and Szegedy, 2015] and LayerNorm [Ba et al., 2016]. Finally, instead of directly applying normalization to the embeddings, we replace the MSE in the encoder loss (Equation 2) by the cosine loss from Schwarzer et al. [2020].

> $\Rightarrow$ The usage of AvgL1Norm is beneficial and related alternate approaches do not achieve the same performance.

**End-to-end.** Embeddings can be trained independently or end-to-end with the downstream task. We test our approach as an auxiliary loss to the value function. The encoders and the value function are trained end-to-end, thus allowing the value loss to affect the embeddings $(z^{sa}, z^s)$, where the encoder loss (Equation 2) is multiplied by a constant to weigh its importance versus the value loss.

> $\Rightarrow$ Learning the embeddings end-to-end with the value function performs signficantly worse than decoupled representation learning.

# 5 Stabilizing RL with Decoupled Representation Learning

In this section, we present the TD7 algorithm (TD3+4 additions). We begin by introducing the use of checkpoints in RL to improve the stability of RL agents. We then combine SALE with checkpoints and various previous algorithmic modifications to TD3 [Fujimoto et al., 2018] to create a single RL algorithm for both the online and offline setting.

## 5.1 Policy Checkpoints

Deep RL algorithms are notoriously unstable [Henderson et al., 2017]. The unreliable nature of deep RL algorithms suggest a need for stabilizing techniques. While we can often directly address the source of instability, some amount of instability is inherent to the combination of function approximation and RL. In this section, we propose the use of checkpoints, to preserve evaluation performance, irrespective of the quality of the current learned policy.

A checkpoint is a snapshot of the parameters of a model, captured at a specific time during training. In supervised learning, checkpoints are often used to recall a previous set of high-performing parameters based on validation error, and maintain a consistent performance across evaluations [Vaswani et al., 2017, Kenton and Toutanova, 2019]. Yet this technique is surprisingly absent from the deep RL toolkit for stabilizing policy performance.

In RL, using the checkpoint of a policy that obtained a high reward during training, instead of the current policy, could improve the stability of the performance at test time.

For off-policy deep RL algorithms, the standard training paradigm is to train after each time step (typically at a one-to-one ratio: one gradient step for one data point). However, this means that the policy changes throughout each episode, making it hard to evaluate the performance. Similar to many on-policy algorithms [Williams, 1992, Schulman et al., 2017], we propose to keep the policy fixed for several *assessment* episodes, then batch the training that would have occurred.

- Standard off-policy RL: Collect a data point → train once.
- Proposed: Collect $N$ data points over several assessment episodes → train $N$ times.

In a similar manner to evolutionary approaches [Salimans et al., 2017], we can use these assessment episodes to judge if the current policy outperforms the previous best policy and checkpoint accordingly. At evaluation time, the checkpoint policy is used, rather than the current policy.

We make two additional modifications to this basic strategy.

**Minimum over mean.** Setting aside practical considerations, the optimal approach would be to evaluate the average performance of each policy using as many trials as possible. However, to preserve learning speed and sample efficiency, it is only sensible to use a handful of trials. As such, to penalize unstable policies using a finite number of assessment episodes, we use the minimum performance, rather than the mean performance. This approach also means that extra assessment episodes do not need to be wasted on poorly performing policies, since training can resume early if the performance of any episode falls below the checkpoint performance.

**Variable assessment length.** In Appendix F, we examine the caliber of policies trained with a varied number of assessment episodes and observe that a surprisingly high number of episodes (20+) can be used without compromising the performance of the final policy. However, the use of many assessment episodes negatively impacts the early performance of the agent. Freezing training for many episodes means that the environment is explored by a stale policy, reducing data diversity, and delaying feedback from policy updates. To counteract this effect, we restrict the number of assessment episodes used during the initial phase of training before increasing it.

Additional details of our approach to policy checkpoints can be found in Appendix F.

## 5.2 TD7

TD7 is based on TD3 [Fujimoto et al., 2018] with LAP [Fujimoto et al., 2020], a behavior cloning term for offline RL [Fujimoto and Gu, 2021], SALE (Section 4.1), and policy checkpoints (Section 5.1).

**LAP.** Gathered experience is stored in a replay buffer [Lin, 1992] and sampled according to LAP [Fujimoto et al., 2020], a prioritized replay buffer $D$ [Schaul et al., 2016] where a transi-

---

**Algorithm 1** Online TD7

---

1: **Initialize:**                                                                                  ▷ *Before training*
    · Policy $\pi_{t+1}$, value function $Q_{t+1}$, encoders $(f_{t+1}, g_{t+1})$.
    · Target policy $\pi_t$, target value function $Q_t$, fixed encoders $(f_t, g_t)$, target fixed encoders $(f_{t-1}, g_{t-1})$.
    · Checkpoint policy $\pi_c$, checkpoint encoder $f_c$.

2: **for** `episode = 1 to final_episode` **do**                                                    ▷ *Data collection*
3:     Using current policy $\pi_{t+1}$, collect transitions and store in the LAP replay buffer.

4:     **if** `checkpoint_condition` **then**                                         ▷ *Checkpointing*
5:         **if** actor $\pi_{t+1}$ outperforms checkpoint policy $\pi_c$ **then**
6:             Update checkpoint networks $\pi_c \leftarrow \pi_{t+1}$, $f_c \leftarrow f_t$.

7:     **for** $i = 1$ **to** `timesteps_since_training` **do**                      ▷ *Training*
8:         Sample transitions from LAP replay buffer (Equation 10).
9:         Train encoder (Equation 2), value function (Equations 6 & 9), and policy (Equation 11).
10:        **if** `target_update_frequency` steps have passed **then**
11:           Update target networks (Equation 8).

    ▷ Detailed hyperparameter explanations found in the Appendix.

---

tion tuple $i := (s, a, r, s')$ is sampled with probability

$$p(i) = \frac{\max\left(|\delta(i)|^{\alpha}, 1\right)}{\sum_{j \in D} \max\left(|\delta(j)|^{\alpha}, 1\right)}, \qquad \text{where } \delta(i) := Q(s, a) - y, \tag{10}$$

where $y$ is the learning target. The amount of prioritization used is controlled by a hyperparameter $\alpha$. Furthermore, the value function loss uses the Huber loss [Huber et al., 1964], rather than the MSE.

**Offline RL.** To make TD7 amenable to the offline RL setting, we add a behavior cloning loss to the policy update [Silver et al., 2014], inspired by TD3+BC [Fujimoto and Gu, 2021]:

$$\pi \approx \arg\max_{\pi} \mathbb{E}_{(s,a) \sim D}\left[Q(s, \pi(s)) - \lambda |\mathbb{E}_{s \sim D}\left[Q(s, \pi(s))\right]|_{\times} \left(\pi(s) - a\right)^2\right]. \tag{11}$$

The same loss function is used for both offline and online RL, where $\lambda = 0$ for the online setting. $|\cdot|_{\times}$ denotes the stop-gradient operation. Unlike TD3+BC, we do not normalize the state vectors. Checkpoints are not used in the offline setting, as there is no interaction with the environment.

Both the value function and policy use the SALE embeddings as input, which we omit from the equations above for simplicity. Pseudocode for TD7 is described in Algorithm 1.

# 6 Results

In this section, we evaluate the performance of TD7 in both the online and offline regimes. A detailed description of the experimental setup, baselines, and hyperparameters can be found in the Appendix, along with additional learning curves and ablation studies.

**Online.** Using OpenAI gym [Brockman et al., 2016], we benchmark TD7 against TD3 [Fujimoto et al., 2018], SAC [Haarnoja et al., 2018], TQC [Kuznetsov et al., 2020], and TD3+OFE [Ota et al., 2020] on the MuJoCo environments [Todorov et al., 2012]. SAC and TD3+OFE results are from re-implementations based on author descriptions [Haarnoja et al., 2018, Ota et al., 2020]. TD3 and TQC results use author-provided code [Fujimoto et al., 2018, Kuznetsov et al., 2020], with a consistent evaluation protocol for all methods. Learning curves are displayed in Figure 4 and final and intermediate results are listed in Table 1.

Although TQC and TD3+OFE use per-environment hyperparameters along with larger and more computationally expensive architectures, TD7 outperforms these baselines significantly in terms of both early (300k time steps) and final performance (5M time steps). At 300k time steps, TD7 often surpasses the performance of TD3 at 5M time steps, highlighting the considerable performance gains.

**Offline.** We benchmark TD7 against CQL [Kumar et al., 2020b], TD3+BC [Fujimoto and Gu, 2021], IQL [Kostrikov et al., 2021] and $\mathcal{X}$-QL [Garg et al., 2023] using the MuJoCo datasets in D4RL [Todorov et al., 2012, Fu et al., 2021]. While there are methods that use per-dataset

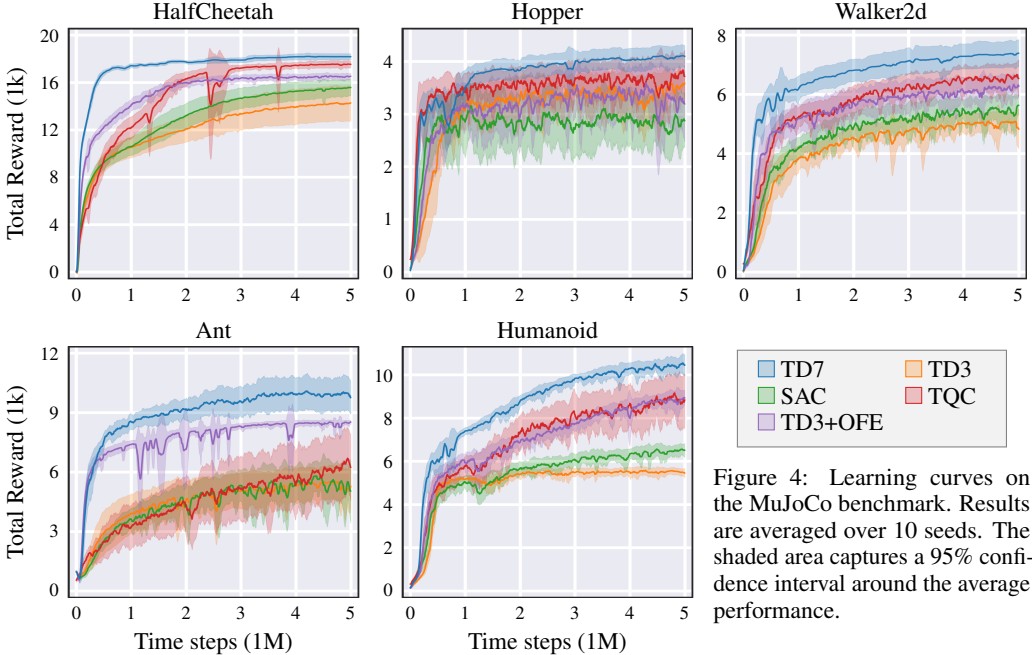

Figure 4: Learning curves on the MuJoCo benchmark. Results are averaged over 10 seeds. The shaded area captures a 95% confidence interval around the average performance.

Table 1: Average performance on the MuJoCo benchmark at 300k, 1M, and 5M time steps, over 10 trials, where $\pm$ captures a 95% confidence interval. The highest performance is highlighted. Any performance which is not statistically significantly worse than the highest performance (according to a Welch's $t$-test with significance level 0.05) is highlighted.

| Environment | Time step | TD3 | SAC | TQC | TD3+OFE | TD7 |
|---|---|---|---|---|---|---|
| HalfCheetah | 300k | $7715 \pm 633$ | $8052 \pm 515$ | $7006 \pm 891$ | $11294 \pm 247$ | $15031 \pm 401$ |
| | 1M | $10574 \pm 897$ | $10484 \pm 659$ | $12349 \pm 878$ | $13758 \pm 544$ | $17434 \pm 155$ |
| | 5M | $14337 \pm 1491$ | $15526 \pm 697$ | $17459 \pm 258$ | $16596 \pm 164$ | $18165 \pm 255$ |
| Hopper | 300k | $1289 \pm 768$ | $2370 \pm 626$ | $3251 \pm 461$ | $1581 \pm 682$ | $2948 \pm 464$ |
| | 1M | $3226 \pm 315$ | $2785 \pm 634$ | $3526 \pm 244$ | $3121 \pm 506$ | $3512 \pm 315$ |
| | 5M | $3682 \pm 83$ | $3167 \pm 485$ | $3462 \pm 818$ | $3423 \pm 584$ | $4075 \pm 225$ |
| Walker2d | 300k | $1101 \pm 386$ | $1989 \pm 500$ | $2812 \pm 838$ | $4018 \pm 570$ | $5379 \pm 328$ |
| | 1M | $3946 \pm 292$ | $4314 \pm 256$ | $5321 \pm 322$ | $5195 \pm 512$ | $6097 \pm 570$ |
| | 5M | $5078 \pm 343$ | $5681 \pm 329$ | $6137 \pm 1194$ | $6379 \pm 332$ | $7397 \pm 454$ |
| Ant | 300k | $1704 \pm 655$ | $1478 \pm 354$ | $1830 \pm 572$ | $6348 \pm 441$ | $6171 \pm 831$ |
| | 1M | $3942 \pm 1030$ | $3681 \pm 506$ | $3582 \pm 1093$ | $7398 \pm 118$ | $8509 \pm 422$ |
| | 5M | $5589 \pm 758$ | $4615 \pm 2022$ | $6329 \pm 1510$ | $8547 \pm 84$ | $10133 \pm 966$ |
| Humanoid | 300k | $1344 \pm 365$ | $1997 \pm 483$ | $3117 \pm 910$ | $3181 \pm 771$ | $5332 \pm 714$ |
| | 1M | $5165 \pm 145$ | $4909 \pm 364$ | $6029 \pm 531$ | $6032 \pm 334$ | $7429 \pm 153$ |
| | 5M | $5433 \pm 245$ | $6555 \pm 279$ | $8361 \pm 1364$ | $8951 \pm 246$ | $10281 \pm 588$ |

hyperparameters to attain higher total results, we omit these methods because it makes it difficult to directly compare. Baseline results are obtained by re-running author-provided code with a single set of hyperparameters and a consistent evaluation protocol. Final performance is reported in Table 2.

TD7 outperforms all baselines. Since TD7 and TD3+BC employ the same approach to offline RL, the significant performance gap highlights the effectiveness of SALE in the offline setting.

**Ablation study**. In Figure 5 we report the results of an ablation study over the components of TD7 (SALE, checkpoints, LAP). The interaction between components is explored further in Appendix G.

**Run time.** To understand the computational cost of using SALE and the TD7 algorithm, we benchmark the run time of each of the online baselines with identical computational resources and deep learning framework. The results are reported in Figure 6.

Table 2: Average final performance on the D4RL benchmark after training for 1M time steps. over 10 trials, where ± captures a 95% confidence interval. The highest performance is highlighted. Any performance which is not statistically significantly worse than the highest performance (according to a Welch's $t$-test with significance level 0.05) is highlighted.

| Environment | Dataset | CQL | TD3+BC | IQL | $\mathcal{X}$-QL | TD7 |
|---|---|---|---|---|---|---|
| HalfCheetah | Medium | $46.7 \pm 0.3$ | $48.1 \pm 0.1$ | $47.4 \pm 0.2$ | $47.4 \pm 0.1$ | $58.0 \pm 0.4$ |
| | Medium-Replay | $45.5 \pm 0.3$ | $44.6 \pm 0.4$ | $43.9 \pm 1.3$ | $44.2 \pm 0.7$ | $53.8 \pm 0.8$ |
| | Medium-Expert | $76.8 \pm 7.4$ | $93.7 \pm 0.9$ | $89.6 \pm 3.5$ | $90.2 \pm 2.7$ | $104.6 \pm 1.6$ |
| Hopper | Medium | $59.3 \pm 3.3$ | $59.1 \pm 3.0$ | $63.9 \pm 4.9$ | $67.7 \pm 3.6$ | $76.1 \pm 5.1$ |
| | Medium-Replay | $78.8 \pm 10.9$ | $52.0 \pm 10.6$ | $93.4 \pm 7.8$ | $82.0 \pm 14.9$ | $91.1 \pm 8.0$ |
| | Medium-Expert | $79.9 \pm 19.8$ | $98.1 \pm 10.7$ | $64.2 \pm 32.0$ | $92.0 \pm 10.0$ | $108.2 \pm 4.8$ |
| Walker2d | Medium | $81.4 \pm 1.7$ | $84.3 \pm 0.8$ | $84.2 \pm 1.6$ | $79.2 \pm 4.0$ | $91.1 \pm 7.8$ |
| | Medium-Replay | $79.9 \pm 3.6$ | $81.0 \pm 3.4$ | $71.2 \pm 8.3$ | $61.8 \pm 7.7$ | $89.7 \pm 4.7$ |
| | Medium-Expert | $108.5 \pm 1.2$ | $110.5 \pm 0.4$ | $108.9 \pm 1.4$ | $110.3 \pm 0.2$ | $111.8 \pm 0.6$ |
| Total | | $656.7 \pm 24.3$ | $671.3 \pm 15.7$ | $666.7 \pm 34.6$ | $674.9 \pm 20.4$ | $784.4 \pm 14.1$ |

# 7 Conclusion

Representation learning has been typically reserved for image-based tasks, where the observations are large and unstructured. However, by learning embeddings which consider the interaction between state and action, we make representation learning more broadly applicable to low-level states. We introduce SALE, a method for learning state-action embeddings by considering a latent space dynamics model. Through an extensive empirical evaluation, we investigate various design choices in SALE.

We highlight the risk of extrapolation error [Fujimoto et al., 2019] due to the increase in input dimensions from using state-action embeddings, but show this instability can be corrected by clipping the target with seen values. We further introduce stability by including policy checkpoints.

While both SALE and policy checkpoints are general-purpose techniques that can be included with most RL methods, we combine them with TD3 and several other recent improvements [Fujimoto et al., 2020, Fujimoto and Gu, 2021] to introduce the TD7 algorithm. We find our TD7 algorithm is able to match the performance of expensive offline algorithms and significantly outperform the state-of-the-art continuous control methods in both final performance and early learning.

## Acknowledgments and Disclosure of Funding

This research was enabled in part by support provided by Calcul Québec and the Digital Research Alliance of Canada.

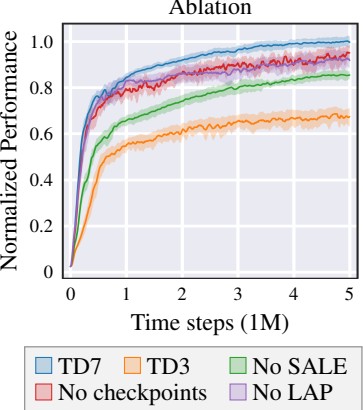

Figure 5: Ablation study over the components of TD7. The y-axis corresponds to the average performance over all five MuJoCo tasks, normalized with respect to the performance of TD7 at 5M time steps. The shaded area captures a 95% confidence interval. The impact can be ranked (1) SALE, (2) LAP, (3) policy checkpoints.

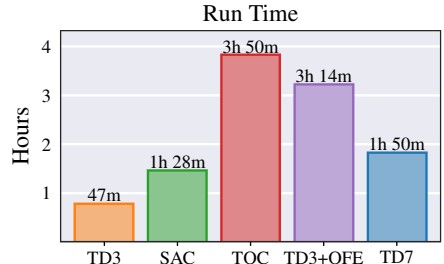

Figure 6: Run time of each method for 1M time steps on the HalfCheetah environment, using the same hardware and deep learning framework (PyTorch [Paszke et al., 2019]).

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

# Appendix

## Table of Contents

# A TD7 Additional Details

## A.1 Algorithm

TD7 (TD3+4 additions) has several networks and sub-components:

- Two value functions $(Q_{t+1,1}, Q_{t+1,2})$.
- Two target value functions $(Q_{t,1}, Q_{t,2})$.
- A policy network $\pi_{t+1}$.
- A target policy network $\pi_t$.
- An encoder, with sub-components $(f_{t+1}, g_{t+1})$.
- A fixed encoder, with sub-components $(f_t, g_t)$.
- A target fixed encoder with sub-components $(f_{t-1}, g_{t-1})$.
- A checkpoint policy $\pi_c$ and checkpoint encoder $f_c$ ($g$ is not needed).

**Encoder:** The encoder is composed of two sub-networks $(f_{t+1}(s), g_{t+1})(z^s, a)$, where each network outputs an embedding:

$$z^s := f(s), \qquad z^{sa} := g(z^s, a). \tag{12}$$

At each training step, the encoder is updated with the following loss:

$$\mathcal{L}(f_{t+1}, g_{t+1}) := \Big( g_{t+1}(f_{t+1}(s), a) - |f_{t+1}(s')|_\times \Big)^2 \tag{13}$$

$$= \Big( z^{sa}_{t+1} - |z^{s'}_{t+1}|_\times \Big)^2, \tag{14}$$

where $|\cdot|_\times$ is the stop-gradient operation.

**Value function:** TD7 uses a pair of value functions (as motivated by TD3 [Fujimoto et al., 2018]) $(Q_{t+1,1}, Q_{t+1,2})$, each taking input $[z^{s'a'}_{t-1}, z^{s'}_{t-1}, s', a']$. At each training step, both value functions are updated with the following loss:

$$\mathcal{L}(Q_{t+1}) := \text{Huber}\Big( \text{target} - Q_{t+1}(z^{sa}_t, z^s_t, s, a) \Big), \tag{15}$$

$$\text{target} := r + \gamma \,\text{clip}\big( \min \left( Q_{t,1}(x), Q_{t,2}(x) \right), Q_{\min}, Q_{\max} \big), \tag{16}$$

$$x := [z^{s'a'}_{t-1}, z^{s'}_{t-1}, s', a'], \tag{17}$$

$$a' := \pi_t(z^{s'}_{t-1}, s') + \epsilon, \tag{18}$$

$$\epsilon \sim \text{clip}(\mathcal{N}(0, \sigma^2), -c, c). \tag{19}$$

Taking the minimum of the value functions is from TD3's Clipped Double Q-learning (CDQ) [Fujimoto et al., 2018]). The use of Huber loss [Huber et al., 1964] is in accordance to TD3 with the Loss-Adjusted Prioritized (LAP) experience replay [Fujimoto et al., 2020]. $a'$ is sampled and clipped in the same manner as TD3 [Fujimoto et al., 2018]. The same embeddings $(z^{sa}_t, z^s_t)$ are used for each value function. $Q_{\min}$ and $Q_{\max}$ are updated at each time step:

$$Q_{\min} \leftarrow \min \left( Q_{\min}, \text{target} \right), \tag{20}$$

$$Q_{\max} \leftarrow \max \left( Q_{\max}, \text{target} \right), \tag{21}$$

where target is defined by Equation 16.

**Policy:** TD7 uses a single policy network which takes input $[z^s, s]$. On every second training step (according to TD3's delayed policy updates [Fujimoto et al., 2018]) the policy $\pi_{t+1}$ is updated with the following loss:

$$\mathcal{L}(\pi_{t+1}) := -Q + \lambda |\mathbb{E}_{s \sim D}[Q]|_\times (a_\pi - a)^2, \tag{22}$$

$$Q := 0.5 \left( Q_{t+1,1}(x) + Q_{t+1,2}(x) \right) \tag{23}$$

$$x := [z^{sa_\pi}_t, z^s_t, s, a_\pi], \tag{24}$$

$$a_\pi := \pi_{t+1}(z^s_t, s). \tag{25}$$

The policy loss is the deterministic policy gradient (DPG) [Silver et al., 2014] with a behavior cloning term to regularize [Fujimoto and Gu, 2021]. $\lambda$ is set to 0 for online RL.

After every `target_update_frequency` (250) training steps, the iteration is updated and each target (and fixed) network copies the network of the higher iteration:

$$(Q_{t,1}, Q_{t,2}) \leftarrow (Q_{t+1,1}, Q_{t+1,2}), \tag{26}$$

$$\pi_t \leftarrow \pi_{t+1}, \tag{27}$$

$$(f_{t-1}, g_{t-1}) \leftarrow (f_t, g_t), \tag{28}$$

$$(f_t, g_t) \leftarrow (f_{t+1}, g_{t+1}). \tag{29}$$

The checkpoint policy and checkpoint encoder are only used at test time (see Appendix F).

**LAP:** Gathered experience is stored in a replay buffer [Lin, 1992] and sampled according to LAP [Fujimoto et al., 2020], a prioritized replay buffer $D$ [Schaul et al., 2016] where a transition tuple $i := (s, a, r, s')$ is sampled with probability

$$p(i) = \frac{\max\left(|\delta(i)|^\alpha, 1\right)}{\sum_{j \in D} \max\left(|\delta(j)|^\alpha, 1\right)}, \tag{30}$$

$$|\delta(i)| := \max\left(\left|Q_{t+1,1}(z_t^{sa}, z_t^s, s, a) - \mathtt{target}\right|, \left|Q_{t+1,2}(z_t^{sa}, z_t^s, s, a) - \mathtt{target}\right|\right), \tag{31}$$

where `target` is defined by Equation 16. As suggested by Fujimoto et al. [2020], $|\delta(i)|$ is defined by the maximum absolute error of both value functions. The amount of prioritization used is controlled by a hyperparameter $\alpha$. New transitions are assigned the maximum priority of any sample in the replay buffer.

We outline the train function of TD7 in Algorithm 2. There is no difference between the train function of online TD7 and offline TD7 other than the value of $\lambda$, which is 0 for online and 0.1 for offline.

---

**Algorithm 2** TD7 Train Function

---

1: Sample transition from LAP replay buffer with probability (Equation 30).
2: Train encoder (Equation 13).
3: Train value function (Equation 15).
4: Update $(Q_{\min}, Q_{\max})$ (Equations 20 & 21).
5: **if** $i$ mod `policy_update_frequency` $= 0$ **then**
6:     Train policy (Equation 22).
7: **if** $i$ mod `target_update_frequency` $= 0$ **then**
8:     Update target networks (Equation 26).

---

## A.2 Hyperparameters

The action space is assumed to be in the range $[-1, 1]$ (and is normalized if otherwise). Besides a few exceptions mentioned below, hyperparameters (and architecture) are taken directly from TD3 https://github.com/sfujim/TD3.

Table 3: TD7 Hyperparameters.

|  | Hyperparameter | Value |
|---|---|---|
| TD3 [Fujimoto et al., 2018] | Target policy noise $\sigma$
Target policy noise clipping $c$
Policy update frequency | $\mathcal{N}(0, 0.2^2)$
$(-0.5, 0.5)$
2 |
| LAP [Fujimoto et al., 2020] | Probability smoothing $\alpha$
Minimum priority | 0.4
1 |
| TD3+BC [Fujimoto and Gu, 2021] | Behavior cloning weight $\lambda$ (Online)
Behavior cloning weight $\lambda$ (Offline) | 0.0
0.1 |
| Policy Checkpoints (Appendix F) | Checkpoint criteria
Early assessment episodes
Late assessment episodes
Early time steps
Criteria reset weight | minimum
1
20
750k
0.9 |
| Exploration | Initial random exploration time steps
Exploration noise | 25k
$\mathcal{N}(0, 0.1^2)$ |
| Common | Discount factor $\gamma$
Replay buffer capacity
Mini-batch size
Target update frequency | 0.99
1M
256
250 |
| Optimizer | (Shared) Optimizer
(Shared) Learning rate | Adam [Kingma and Ba, 2014]
$3e - 4$ |

Besides algorithmic difference from TD3, there are three implementation-level changes:

1. Rather than only using $Q_1$, both value functions are used when updating the policy (Equation 22).
2. The value function uses ELU activation functions [Clevert et al., 2015] rather than ReLU activation functions.
3. The target network is updated periodically (every 250 time steps) rather than using an exponential moving average at every time step. This change is necessary due to the use of fixed encoders.

We evaluate the importance of these changes in our ablation study (Appendix G). Network architecture details are described in Pseudocode 1.

## Pseudocode 1. TD7 Network Details

**Variables:**

```
zs_dim = 256
```

**Value $Q$ Network:**

▷ TD7 uses two value networks each with the same network and forward pass.

```
l0 = Linear(state_dim + action_dim, 256)
l1 = Linear(zs_dim * 2 + 256, 256)
l2 = Linear(256, 256)
l3 = Linear(256, 1)
```

**Value $Q$ Forward Pass:**

```
input = concatenate([state, action])
x = AvgL1Norm(l0(inuput))
x = concatenate([zsa, zs, x])
x = ELU(l1(x))
x = ELU(l2(x))
value = l3(x)
```

**Policy $\pi$ Network:**

```
l0 = Linear(state_dim, 256)
l1 = Linear(zs_dim + 256, 256)
l2 = Linear(256, 256)
l3 = Linear(256, action_dim)
```

**Policy $\pi$ Forward Pass:**

```
input = state
x = AvgL1Norm(l0(input))
x = concatenate([zs, x])
x = ReLU(l1(x))
x = ReLU(l2(x))
action = tanh(l3(x))
```

**State Encoder $f$ Network:**

```
l1 = Linear(state_dim, 256)
l2 = Linear(256, 256)
l3 = Linear(256, zs_dim)
```

**State Encoder $f$ Forward Pass:**

```
input = state
x = ELU(l1(input))
x = ELU(l2(x))
zs = AvgL1Norm(l3(x))
```

**State-Action Encoder $g$ Network:**

```
l1 = Linear(action_dim + zs_dim, 256)
l2 = Linear(256, 256)
l3 = Linear(256, zs_dim)
```

**State-Action Encoder $g$ Forward Pass:**

```
input = concatenate([action, zs])
x = ELU(l1(input))
x = ELU(l2(x))
zsa = l3(x)
```

# B  Experimental Details

**Environment.** Our experimental evaluation is based on the MuJoCo simulator [Todorov et al., 2012] with tasks defined by OpenAI gym [Brockman et al., 2016] using the v4 environments. No modification are made to the state, action, or reward space.

**Terminal transitions.** All methods use a discount factor $\gamma = 0.99$ for non-terminal transitions and $\gamma = 0$ for terminal transitions. The final transition from an episode which ends due to a time limit is not considered terminal.

**Exploration.** To fill the replay buffer before training, all methods initially collect data by following a uniformly random policy for the first 25k time steps (256 for TQC). This initial data collection is accounted for in all graphs and tables, meaning "time steps" refers to the number of environment interactions, rather than the number of training steps. Methods based on the deterministic TD3 add Gaussian noise to the policy. Methods based on the stochastic SAC do not add noise.

**Evaluation.** Agents are evaluated every 5000 time steps, taking the average undiscounted sum of rewards over 10 episodes. Each experiment is repeated over 10 seeds. Evaluations use a deterministic policy, meaning there is no noise added to the policy and stochastic methods use the mean action. Methods with our proposed policy checkpoints use the checkpoint policy during evaluations. Evaluations are considered entirely independent from training, meaning no data is saved, nor is any network updated.

**Visualization and tables.** As aforementioned, time steps in figures and tables refers to the number of environment interactions (all methods are trained once per environment interaction, other than the initial random data collection phase). The value reported by the table corresponds to the average evaluation over the 10 seeds, at the given time step (300k, 1M or 5M for online results and 1M for offline results). The shaded area in figures and the $\pm$ term in tables refers to a $97.5\%$ confidence interval. Given that methods are evaluated over 10 seeds, this confidence interval $\texttt{CI}$ is computed by

$$\texttt{CI} = \frac{1.96}{\sqrt{10}}\sigma, \tag{32}$$

where $\sigma$ is the sample standard deviation with Bessel's correction at the corresponding evaluation. Unless stated otherwise, all curves are smoothed uniformly over a window of 10 evaluations.

**Offline Experiments.** Offline results are based on the D4RL datasets [Fu et al., 2021], using the v2 version. No modification are made to the state, action, or reward space. The reported performance is from a final evaluation occurring after 1M training steps, which uses the average D4RL score over 10 episodes. TD7 results are repeated over 10 seeds. Baseline results are taken from other papers, and may not be based on an identical evaluation protocol in terms of number of episodes and seeds used.

**Software.** We use the following software versions:

- Python 3.9.13
- Pytorch 2.0.0 [Paszke et al., 2019]
- CUDA version 11.8
- Gym 0.25.0 [Brockman et al., 2016]
- MuJoCo 2.3.3 [Todorov et al., 2012]

# C   Baselines

## C.1   TD3 & TD3+OFE Hyperparameters

Our TD3 baseline [Fujimoto et al., 2018] uses the author implementation `https://github.com/sfujim/TD3`. Our TD3+OFE baseline [Ota et al., 2020] uses the aforementioned TD3 code alongside a re-implementation of OFENet. For hyperparameters, the action space is assumed to be in the range $[-1, 1]$.

**Environment-specific hyperparameters.** OFE uses a variable number of encoder layers depending on the environment. However, the authors also provide an offline approach for hyperparameter tuning based on the representation loss. Our choice of layers per environment is the outcome of their hyperparameter tuning. Additionally, when training the state-action encoder, OFE drops certain state dimensions which correspond to external forces which are hard to predict. In practice this only affects Humanoid, which uses 292 out of a possible state 376 dimensions on Humanoid. All other tasks use the full state space.

Hyperparameters of both methods are listed in Table 4. Network architecture details of TD3 are described in Pseudocode 2. Network architecture details of TD3+OFE are described in Pseudocode 3 and Pseudocode 4.

Table 4: TD3 & TD3+OFE Hyperparameters.

|  | Hyperparameter | Value |
|---|---|---|
|  | Encoder layers |  |
| OFE [Ota et al., 2020] | HalfCheetah | 8 |
|  | Hopper | 6 |
|  | Walker2d | 6 |
|  | Ant | 6 |
|  | Humanoid | 8 |
| TD3 [Fujimoto et al., 2018] | Target policy noise $\sigma$ | $\mathcal{N}(0, 0.2^2)$ |
|  | Target policy noise clipping $c$ | $(-0.5, 0.5)$ |
|  | Policy update frequency | 2 |
| Exploration | Initial random exploration time steps | 25k |
|  | Exploration noise | $\mathcal{N}(0, 0.1^2)$ |
| Common | Discount factor $\gamma$ | 0.99 |
|  | Replay buffer capacity | 1M |
|  | Mini-batch size | 256 |
|  | Target update rate $\tau$ | 0.005 |
| Optimizer | (Shared) Optimizer | Adam [Kingma and Ba, 2014] |
|  | (Shared) Learning rate | $3e-4$ |

## Pseudocode 2. TD3 Network Details

**Value $Q$ Network:**
▷ TD3 uses two value networks each with the same network and forward pass.

```
l1 = Linear(state_dim + action_dim, 256)
l2 = Linear(256, 256)
l3 = Linear(256, 1)
```

**Value $Q$ Forward Pass:**

```
input = concatenate([state, action])
x = ReLU(l1(input))
x = ReLU(l2(x))
value = l3(x)
```

**Policy $\pi$ Network:**

```
l1 = Linear(state_dim, 256)
l2 = Linear(256, 256)
l3 = Linear(256, action_dim)
```

**Policy $\pi$ Forward Pass:**

```
input = state
x = ReLU(l1(input))
x = ReLU(l2(x))
action = tanh(l3(x))
```

## Pseudocode 3. TD3+OFE Network Details (TD3)

**Variables:**

```
zs_dim = 240
zsa_dim = 240
num_layers = 6 (or 8)
zs_hdim = zs_dim/num_layers = 40 (or 30)
zsa_hdim = zsa_dim/num_layers = 40 (or 30)
target_dim = state_dim if not Humanoid else 292
```

**Value $Q$ Network:**
▷ TD3+OFE uses two value networks each with the same network and forward pass.

```
l1 = Linear(state_dim + action_dim + zs_dim + zsa_dim, 256)
l2 = Linear(256, 256)
l3 = Linear(256, 1)
```

**Value $Q$ Forward Pass:**

```
input = zsa
x = ReLU(l1(input))
x = ReLU(l2(x))
value = l3(x)
```

**Policy $\pi$ Network:**

```
l1 = Linear(state_dim + zs_dim, 256)
l2 = Linear(256, 256)
l3 = Linear(256, action_dim)
```

**Policy $\pi$ Forward Pass:**

```
input = zs
x = ReLU(l1(input))
x = ReLU(l2(x))
action = tanh(l3(x))
```

**Pseudocode 4. TD3+OFE Network Details (OFE)**

**State Encoder $f^s$ Network:**

```
l1 = Linear(state_dim, zs_hdim)
l2 = Linear(state_dim + zs_hdim, zs_hdim)
l3 = Linear(state_dim + zs_hdim * 2, zs_hdim)
...
l_num_layers = Linear(state_dim + zs_hdim * (num_layers-1), zs_hdim)
```

**State Encoder $f^s$ Forward Pass:**

```
input = state
x = swish(batchnorm(l1(input)))
input = concatenate([input, x])
x = swish(batchnorm(l2(input)))
...
x = swish(batchnorm(l_num_layers(input)))
output = concatenate([input, x])
```

**State-Action Encoder $f^{sa}$ Network:**

```
l1 = Linear(state_dim + action_dim + zs_dim, zsa_hdim)
l2 = Linear(state_dim + action_dim + zs_dim + zsa_hdim, zsa_hdim)
l3 = Linear(state_dim + action_dim + zs_dim + zsa_hdim * 2, zsa_hdim)
...
l_num_layers = Linear(state_dim + action_dim
    + zs_dim + zsa_hdim * (num_layers-1), zsa_hdim)

final_layer = Linear(state_dim + action_dim + zs_dim + zsa_dim, target_dim)
```

**State-Action Encoder $f^{sa}$ Forward Pass:**

```
input = concatenate([action, zs])
x = swish(batchnorm(l1(input)))
input = concatenate([input, x])
x = swish(batchnorm(l2(input)))
...
x = swish(batchnorm(l_num_layers(input)))
output = concatenate([input, x])
```

**Final Layer $t$ Forward Pass:**

```
input = zsa
output = final_layer(zsa)
```

## C.2 SAC & TQC Hyperparameters

Our SAC baseline [Haarnoja et al., 2018] is based on our TD3 implementation, keeping hyper-parameters constant when possible. Remaining details are based on the author implementation https://github.com/haarnoja/sac. Our TQC baseline [Kuznetsov et al., 2020] uses the author PyTorch implementation https://github.com/SamsungLabs/tqc_pytorch (with evaluation code kept consistent for all methods). For hyperparameters, the action space is assumed to be in the range $[-1, 1]$.

TQC is based on SAC and uses similar hyperparameters. One exception is the use of an $\epsilon$ offset. As suggested in the appendix of Haarnoja et al. [2018], the log probability of the Tanh Normal distribution is calculated as follows:

$$\log \pi(a|s) = \log N(u|s) - \log(1 - \tanh(u)^2 + \epsilon), \tag{33}$$

where $u$ is the pre-activation value of the action, sampled from $N$, where $N$ is distributed according to $\mathcal{N}(\mu, \sigma^2)$, from the outputs $\mu$ and $\log \sigma$ of the actor network. Kuznetsov et al. [2020] use an alternate calculation of the log of the Tanh Normal distribution which eliminates the need for an $\epsilon$ offset:

$$\log \pi(a|s) = \log N(u|s) - (2\log(2) + \log(\mathrm{sigmoid}(2u)) + \log(\mathrm{sigmoid}(-2u))). \tag{34}$$

**Environment-specific hyperparameters.** TQC varies the number of dropped quantiles depending on the environment. While it is possible to select this quantity based on heuristics [Kuznetsov et al., 2021], we use the author suggested values for each environment.

Hyperparameters of SAC are listed in Table 5. Network architecture details of SAC are described in Pseudocode 5.

Hyperparameters of TQC are listed in Table 6. Network architecture details of TQC are described in Pseudocode 6.

Table 5: SAC Hyperparameters.

| | Hyperparameter | Value |
|---|---|---|
| SAC [Haarnoja et al., 2018] | Target Entropy
Policy $\log$ standard deviation clamp
Numerical stability offset $\epsilon$ | $-$action_dim
$[-20, 2]$
$1e-6$ |
| Exploration | Initial random exploration time steps | 25k |
| Common | Discount factor $\gamma$
Replay buffer capacity
Mini-batch size
Target update rate $\tau$ | 0.99
1M
256
0.005 |
| Optimizer | (Shared) Optimizer
(Shared) Learning rate | Adam [Kingma and Ba, 2014]
$3e-4$ |

---

**Pseudocode 5. SAC Network Details**

**Value $Q$ Network:**
▷ SAC uses two value networks each with the same network and forward pass.

```
l1 = Linear(state_dim + action_dim, 256)
l2 = Linear(256, 256)
l3 = Linear(256, 1)
```

**Value $Q$ Forward Pass:**

```
input = concatenate([state, action])
x = ReLU(l1(input))
x = ReLU(l2(x))
value = l3(x)
```

---

**Policy $\pi$ Network:**

```
l1 = Linear(state_dim, 256)
l2 = Linear(256, 256)
l3 = Linear(256, action_dim * 2)
```

**Policy $\pi$ Forward Pass:**

```
input = state
x = ReLU(l1(input))
x = ReLU(l2(x))
mean, log_std = l3(x)
x = Normal(mean, exp(log_std.clip(-20, 2))).sample()
action = tanh(x)
```

Table 6: TQC Hyperparameters.

| | Hyperparameter | Value |
|---|---|---|
| | Number of networks | 5 |
| | Quantiles per network | 25 |
| TQC | Quantiles dropped per network | |
| [Kuznetsov et al., 2020] | HalfCheetah | 0 |
| | Hopper | 5 |
| | Walker2d | 2 |
| | Ant | 2 |
| | Humanoid | 2 |
| SAC | Target Entropy | $-$action_dim |
| [Haarnoja et al., 2018] | Policy $\log$ standard deviation clamp | $[-20, 2]$ |
| Exploration | Initial random exploration time steps | 256 |
| | Discount factor $\gamma$ | 0.99 |
| Common | Replay buffer capacity | 1M |
| | Mini-batch size | 256 |
| | Target update rate $\tau$ | 0.005 |
| Optimizer | (Shared) Optimizer | Adam [Kingma and Ba, 2014] |
| | (Shared) Learning rate | $3e-4$ |

---

**Pseudocode 6. TQC Network Details**

**Value $Q$ Network:**
▷ TQC uses five value networks each with the same network and forward pass.

```
l1 = Linear(state_dim + action_dim, 512)
l2 = Linear(512, 512)
l3 = Linear(512, 512)
l4 = Linear(512, 25)
```

**Value $Q$ Forward Pass:**

```
input = concatenate([state, action])
x = ReLU(l1(input))
x = ReLU(l2(x))
x = ReLU(l3(x))
value = l4(x)
```

---

**Policy $\pi$ Network:**

```
l1 = Linear(state_dim, 256)
l2 = Linear(256, 256)
l3 = Linear(256, action_dim * 2)
```

**Policy $\pi$ Forward Pass:**

```
input = state
x = ReLU(l1(input))
x = ReLU(l2(x))
mean, log_std = l3(x)
x = Normal(mean, exp(log_std.clip(-20, 2))).sample()
action = tanh(x)
```

## C.3 Offline RL Baselines

We use four offline RL baseline methods. The results of each method are obtained by re-running the author-provided code with the following commands:

**CQL** [Kumar et al., 2020b]. https://github.com/aviralkumar2907/CQL commit d67dbe9

```
python examples/cql_mujoco_new.py --env=ENV \
--policy_lr=3e-5 --langrange_thresh=-1 --min_q_weight=10
```

Setting the Lagrange threshold below 0 means the Lagrange version of the code is not used (as dictated by the settings defined in the CQL paper). Default hyperparameters in the GitHub performed substantially worse (total normalized score: $221.3 \pm 34.4$). Some modifications were made to the evaluation code to match the evaluation protocol used by TD7.

**TD3+BC** [Fujimoto and Gu, 2021]. https://github.com/sfujim/TD3_BC commit 8791ad7

```
python main.py --env=ENV
```

**IQL** [Kostrikov et al., 2021]. https://github.com/ikostrikov/implicit_q_learning commit 09d7002

```
python train_offline.py --env_name=ENV --config=configs/mujoco_config.py
```

$\mathcal{X}$-**QL** [Garg et al., 2023]. https://github.com/Div99/XQL commit dff09af

```
python train_offline.py --env_name=ENV --config=configs/mujoco_config.py \
--max_clip=7 --double=True --temp=2 --batch_size=256
```

This setting corresponds to the $\mathcal{X}$-QL-C version of the method where hyperparameters are fixed across datasets.

# D  Design Study

In this section we analyze the space of design choices for SALE. All results are based on modifying TD7 without policy checkpoints. A summary of all results is contained in Table 7. Each choice is described in detail in the subsections that follow.

Table 7: Average performance on the MuJoCo benchmark at 1M time steps. $\pm$ captures a 95% confidence interval around the average performance. Results are over 10 seeds.

| Algorithm | HalfCheetah | Hopper | Walker2d | Ant | Humanoid |
|---|---|---|---|---|---|
| TD7 (no checkpoints) | $17123 \pm 296$ | $3361 \pm 429$ | $5718 \pm 308$ | $8605 \pm 1008$ | $7381 \pm 172$ |
| TD3 | $10574 \pm 897$ | $3226 \pm 315$ | $3946 \pm 292$ | $3942 \pm 1030$ | $5165 \pm 145$ |
| Learning Target (Section D.1) | | | | | |
| $s'$ | $16438 \pm 219$ | $3507 \pm 313$ | $5882 \pm 262$ | $7415 \pm 753$ | $6337 \pm 387$ |
| $z_{\text{target}}^{s'}$ | $17317 \pm 65$ | $3034 \pm 647$ | $6227 \pm 218$ | $7939 \pm 334$ | $6984 \pm 578$ |
| $z^s$ and $r$ | $17089 \pm 219$ | $3264 \pm 670$ | $6255 \pm 220$ | $8300 \pm 675$ | $6841 \pm 252$ |
| $z^{s'a'}$ | $12445 \pm 417$ | $3009 \pm 528$ | $4757 \pm 352$ | $6288 \pm 570$ | $5657 \pm 110$ |
| Network Input (Section D.2) | | | | | |
| $Q$, remove $z^{sa}$ | $16948 \pm 250$ | $2914 \pm 756$ | $5746 \pm 254$ | $6911 \pm 552$ | $6907 \pm 443$ |
| $Q$, remove $z^s$ | $16856 \pm 241$ | $2783 \pm 710$ | $5987 \pm 407$ | $7954 \pm 248$ | $7196 \pm 363$ |
| $Q$, remove $s, a$ | $17042 \pm 208$ | $2268 \pm 723$ | $5486 \pm 466$ | $8546 \pm 655$ | $6449 \pm 478$ |
| $Q$, $z^{sa}$ only | $15410 \pm 691$ | $2781 \pm 585$ | $5205 \pm 409$ | $7916 \pm 976$ | $5551 \pm 1144$ |
| $Q$, $s, a$ only | $12776 \pm 774$ | $2929 \pm 631$ | $4938 \pm 422$ | $5695 \pm 875$ | $6209 \pm 384$ |
| $\pi$, $s$ only | $16997 \pm 265$ | $3206 \pm 419$ | $5744 \pm 430$ | $7710 \pm 593$ | $6413 \pm 1521$ |
| $\pi$, $z^s$ only | $17283 \pm 388$ | $2429 \pm 705$ | $6319 \pm 331$ | $8378 \pm 732$ | $6507 \pm 496$ |
| No fixed embeddings | $17116 \pm 149$ | $2262 \pm 590$ | $5835 \pm 481$ | $8027 \pm 551$ | $7334 \pm 153$ |
| Normalization (Section D.3) | | | | | |
| No normalization on $\phi$ | $17231 \pm 246$ | $2647 \pm 466$ | $5639 \pm 1248$ | $8191 \pm 846$ | $5862 \pm 1471$ |
| No normalization | $17275 \pm 288$ | $3359 \pm 479$ | $6168 \pm 164$ | $7274 \pm 662$ | $4803 \pm 1706$ |
| Normalization on $z^{sa}$ | $16947 \pm 284$ | $3383 \pm 262$ | $5502 \pm 1142$ | $8049 \pm 538$ | $6418 \pm 302$ |
| BatchNorm | $17299 \pm 218$ | $2318 \pm 473$ | $4839 \pm 987$ | $7636 \pm 769$ | $6979 \pm 325$ |
| LayerNorm | $17132 \pm 360$ | $2451 \pm 902$ | $4470 \pm 1096$ | $6331 \pm 1024$ | $6712 \pm 1297$ |
| Cosine similarity loss | $16897 \pm 372$ | $3324 \pm 249$ | $5566 \pm 353$ | $7873 \pm 511$ | $4370 \pm 1573$ |
| End-to-end (Section D.4) | | | | | |
| End-to-end, 0.1 | $16186 \pm 360$ | $1820 \pm 674$ | $5013 \pm 729$ | $6601 \pm 1251$ | $5076 \pm 897$ |
| End-to-end, 1 | $15775 \pm 658$ | $1779 \pm 537$ | $4882 \pm 710$ | $6604 \pm 1135$ | $5880 \pm 703$ |
| End-to-end, 10 | $16472 \pm 365$ | $1534 \pm 341$ | $4900 \pm 760$ | $6626 \pm 1253$ | $5279 \pm 1095$ |

### D.1 Learning Target

The encoder $(f_{t+1}, g_{t+1})$ in TD7 uses the following update:

$$\mathcal{L}(f_{t+1}, g_{t+1}) := \left( g_{t+1}(f_{t+1}(s), a) - |f_{t+1}(s')|_{\times} \right)^2 \tag{35}$$

$$= \left( z_{t+1}^{sa} - |z_{t+1}^{s'}|_{\times} \right)^2. \tag{36}$$

In this section, we vary the learning target (originally $z_{t+1}^{s'}$).

([Figure 7](#)) **Target 1:** $s'$. Sets the learning target to the next state $s'$, making the encoder loss:

$$\mathcal{L}(f_{t+1}, g_{t+1}) := \left( z_{t+1}^{sa} - s' \right)^2. \tag{37}$$

This target is inspired by OFENet [Ota et al., 2020].

([Figure 7](#)) **Target 2:** $z_{\text{target}}^{s'}$. Sets the learning target to the embedding of the next state $z_{\text{target}}^{s'}$ taken from a slow-moving target network. This makes the encoder loss:

$$\mathcal{L}(f_{t+1}, g_{t+1}) := \left( z_{t+1}^{sa} - |z_{t+1}^{sa}|_{\times} \right)^2. \tag{38}$$

$z_{\text{target}}^{s'}$ is taken from a target network $f_{\text{target}}$ where $f_{t+1}$ has parameters $\theta$ and $f_{\text{target}}$ has parameters $\theta'$, and $\theta'$ is updated at each time step by:

$$\theta' \leftarrow (0.99)\theta' + (0.01)\theta. \tag{39}$$

This target is inspired by SPR [Schwarzer et al., 2020].

([Figure 7](#)) **Target 3:** $z^s$ and $r$. Sets the learning target to the embedding of the next state $z_{t+1}^s$ (which is the same as in the original loss) but adds a second loss towards the target $r$. This is achieved by having the output of $g$ be two dimensional, with the first dimension being $z_{t+1}^{sa}$ and the second $r_{t+1}^{\text{pred}}$, making the encoder loss:

$$\mathcal{L}(f_{t+1}, g_{t+1}) := \left( g_{t+1}(f_{t+1}(s), a)[0] - |f_{t+1}(s')|_{\times} \right)^2 + \left( g_{t+1}(f_{t+1}(s), a)[1] - r \right)^2 \tag{40}$$

$$= \left( z_{t+1}^{sa} - |z_{t+1}^{s'}|_{\times} \right)^2 + \left( r_{t+1}^{\text{pred}} - r \right)^2. \tag{41}$$

This target is inspired by DeepMDP [Gelada et al., 2019].

([Figure 8](#)) **Target 4:** $z^{s'a'}$. Sets the learning target to the next state-action embedding $z_{t+1}^{s'a'}$, where the action $a'$ is sampled according to the target policy with added noise and clipping (as done in TD3 [Fujimoto et al., 2018]). This makes the encoder loss:

$$\mathcal{L}(f_{t+1}, g_{t+1}) := \left( z_{t+1}^{sa} - |z_{t+1}^{s'a'}|_{\times} \right)^2, \tag{42}$$

$$a' := \pi_t(z_{t-1}^{s'}, s') + \epsilon, \tag{43}$$

$$\epsilon \sim \text{clip}(\mathcal{N}(0, \sigma^2), -c, c). \tag{44}$$

Table 8: Average performance on the MuJoCo benchmark at 1M time steps. $\pm$ captures a $95\%$ confidence interval around the average performance. Results are over 10 seeds.

| Algorithm | HalfCheetah | Hopper | Walker2d | Ant | Humanoid |
|---|---|---|---|---|---|
| TD7 (no checkpoints) | $17123 \pm 296$ | $3361 \pm 429$ | $5718 \pm 308$ | $8605 \pm 1008$ | $7381 \pm 172$ |
| TD3 | $10574 \pm 897$ | $3226 \pm 315$ | $3946 \pm 292$ | $3942 \pm 1030$ | $5165 \pm 145$ |
| $s'$ | $16438 \pm 219$ | $3507 \pm 313$ | $5882 \pm 262$ | $7415 \pm 753$ | $6337 \pm 387$ |
| $z^{s'}_{\text{target}}$ | $17317 \pm 65$ | $3034 \pm 647$ | $6227 \pm 218$ | $7939 \pm 334$ | $6984 \pm 578$ |
| $z^s$ and $r$ | $17089 \pm 219$ | $3264 \pm 670$ | $6255 \pm 220$ | $8300 \pm 675$ | $6841 \pm 252$ |
| $z^{s'a'}$ | $12445 \pm 417$ | $3009 \pm 528$ | $4757 \pm 352$ | $6288 \pm 570$ | $5657 \pm 110$ |

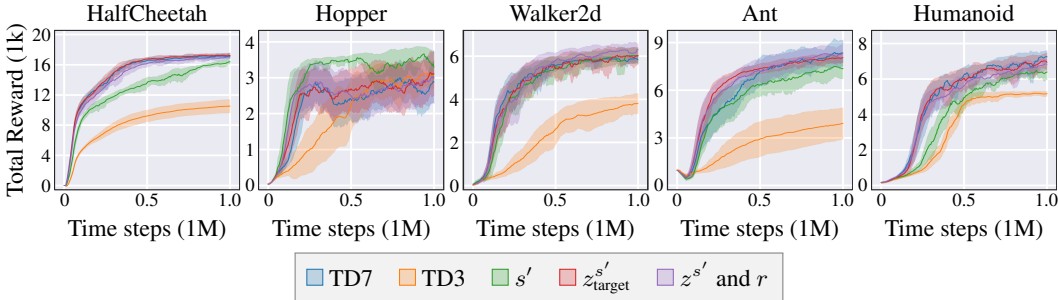

Figure 7: **Learning targets from baseline methods.** Learning curves on the MuJoCo benchmark, using learning targets inspired by baseline methods, the next state $s'$ from OFENet [Ota et al., 2020], the next state embedding $z^{s'}_{\text{target}}$ from a slow-moving target network from SPR [Schwarzer et al., 2020], and including the reward term ($z^{s'}$ and $r$) as in DeepMDP [Gelada et al., 2019]. Note that this study only accounts for changes to the learning target and does not encompass all aspects of the baseline algorithms. Results are averaged over 10 seeds. The shaded area captures a $95\%$ confidence interval around the average performance.

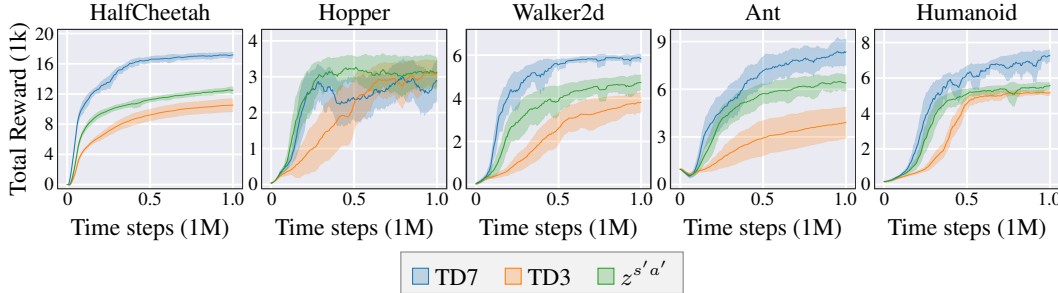

Figure 8: **Alternate learning target.** Learning curves on the MuJoCo benchmark, using a learning target of the next state-action embedding $z^{s'a'}$. Results are averaged over 10 seeds. The shaded area captures a $95\%$ confidence interval around the average performance.

## D.2 Network Input

SALE modifies the value function and policy to take the embeddings as additional input:

$$Q_{t+1}(s,a) \to Q_{t+1}(z_t^{sa}, z_t^s, s, a), \qquad \pi_{t+1}(s) \to \pi_{t+1}(z^s, s). \qquad (45)$$

In this section, we vary the input of the value function and policy. In all cases, changes are made independently (i.e. we fix the policy to the above (using embeddings) and change the policy, or fix the value function to the above (using embeddings) and change the policy).

(Figure 9) **Input 1:** $Q$, remove $z^{sa}$. We remove $z^{sa}$ from the value function input. This makes the input: $Q_{t+1}(z_t^s, s, a)$.

(Figure 9) **Input 2:** $Q$, remove $z^s$. We remove $z^s$ from the value function input. This makes the input: $Q_{t+1}(z_t^{sa}, s, a)$.

(Figure 9) **Input 3:** $Q$, remove $s, a$. We remove $s, a$ from the value function input. This makes the input: $Q_{t+1}(z_t^{sa}, z^s)$.

(Figure 10) **Input 4:** $Q$, $z^{sa}$ only. We only use $z^{sa}$ in the value function input. This makes the input: $Q_{t+1}(z_t^{sa})$.

(Figure 10) **Input 5:** $Q$, $s, a$ only. We only use $s, a$ in the value function input. This makes the input: $Q_{t+1}(s,a)$. As for all cases, the policy still takes in the state embedding $z^s$.

(Figure 11) **Input 6:** $\pi$, $s$ only. We remove $z^s$ from the policy input, using only the state $s$ as input. This makes the input: $\pi_{t+1}(s)$. Note that the value function still uses the embeddings as input.

(Figure 11) **Input 7:** $\pi$, $z^s$ only. We remove $s$ from the policy input, using only the embedding $z^s$ as input. This makes the input: $\pi_{t+1}(z_t^s)$.

(Figure 12) **Input 8:** No fixed embeddings. We remove the fixed embeddings from the value function and the policy. This means that the networks use embeddings from the current encoder, rather than the fixed encoder from the previous iteration. This makes the input: $Q_{t+1}(z_{t+1}^{sa}, z_{t+1}^s, s, a)$ and $\pi_{t+1}(z_{t+1}^s, s)$.

Table 9: Average performance on the MuJoCo benchmark at 1M time steps. $\pm$ captures a 95% confidence interval around the average performance. Results are over 10 seeds.

| Algorithm | HalfCheetah | Hopper | Walker2d | Ant | Humanoid |
|---|---|---|---|---|---|
| TD7 (no checkpoints) | $17123 \pm 296$ | $3361 \pm 429$ | $5718 \pm 308$ | $8605 \pm 1008$ | $7381 \pm 172$ |
| TD3 | $10574 \pm 897$ | $3226 \pm 315$ | $3946 \pm 292$ | $3942 \pm 1030$ | $5165 \pm 145$ |
| $Q$, remove $z^{sa}$ | $16948 \pm 250$ | $2914 \pm 756$ | $5746 \pm 254$ | $6911 \pm 552$ | $6907 \pm 443$ |
| $Q$, remove $z^s$ | $16856 \pm 241$ | $2783 \pm 710$ | $5987 \pm 407$ | $7954 \pm 248$ | $7196 \pm 363$ |
| $Q$, remove $s, a$ | $17042 \pm 208$ | $2268 \pm 723$ | $5486 \pm 466$ | $8546 \pm 655$ | $6449 \pm 478$ |
| $Q$, $z^{sa}$ only | $15410 \pm 691$ | $2781 \pm 585$ | $5205 \pm 409$ | $7916 \pm 976$ | $5551 \pm 1144$ |
| $Q$, $s, a$ only | $12776 \pm 774$ | $2929 \pm 631$ | $4938 \pm 422$ | $5695 \pm 875$ | $6209 \pm 384$ |
| $\pi$, $s$ only | $16997 \pm 265$ | $3206 \pm 419$ | $5744 \pm 430$ | $7710 \pm 593$ | $6413 \pm 1521$ |
| $\pi$, $z^s$ only | $17283 \pm 388$ | $2429 \pm 705$ | $6319 \pm 331$ | $8378 \pm 732$ | $6507 \pm 496$ |
| No fixed embeddings | $17116 \pm 149$ | $2262 \pm 590$ | $5835 \pm 481$ | $8027 \pm 551$ | $7334 \pm 153$ |

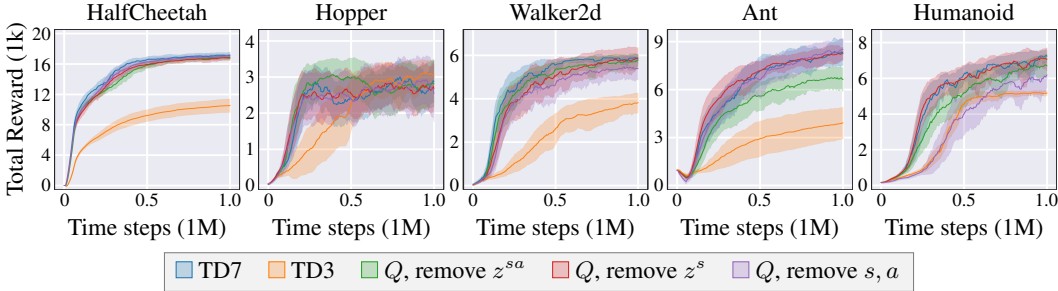

Figure 9: **Removing terms from the value function input.** Learning curves on the MuJoCo benchmark, when removing components from the input to the value function. Results are averaged over 10 seeds. The shaded area captures a 95% confidence interval around the average performance.

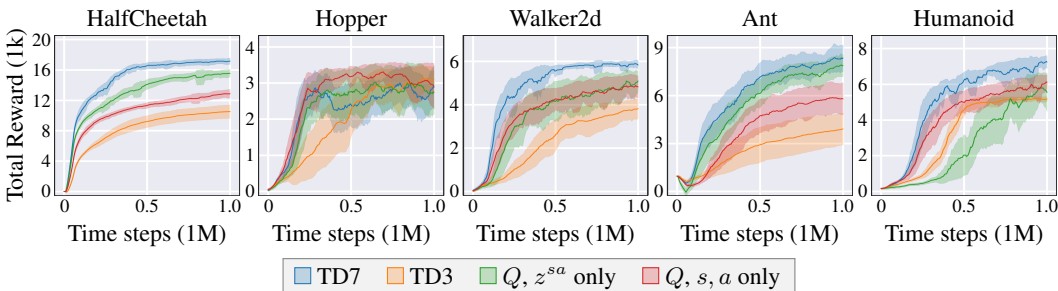

Figure 10: **Exclusive value function input.** Learning curves on the MuJoCo benchmark, changing the value function input to a single component. Results are averaged over 10 seeds. The shaded area captures a 95% confidence interval around the average performance.

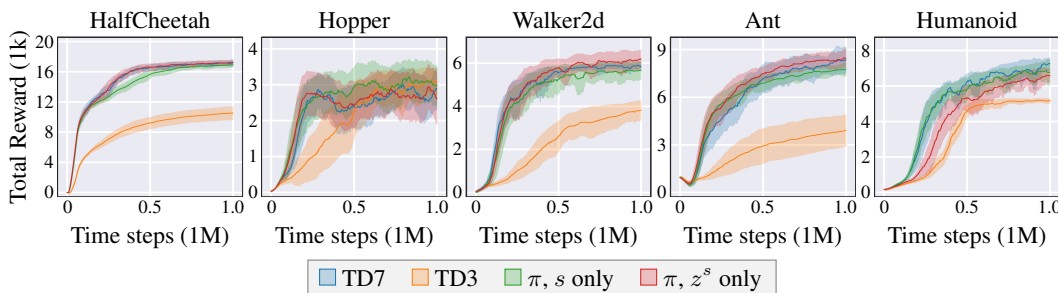

Figure 11: **Modified policy input.** Learning curves on the MuJoCo benchmark, changing the policy input to a single component. Results are averaged over 10 seeds. The shaded area captures a 95% confidence interval around the average performance.

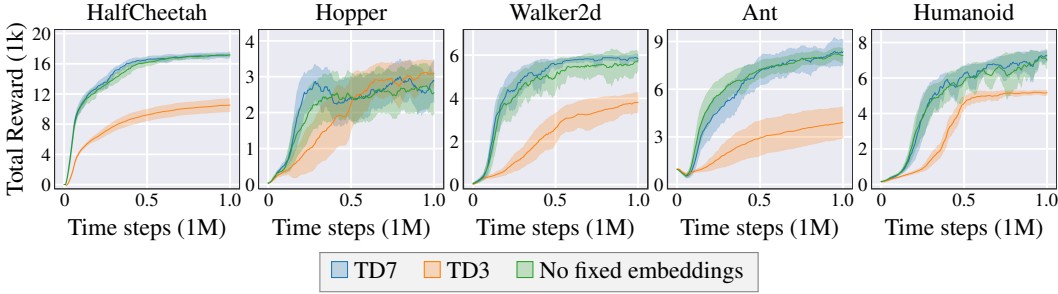

Figure 12: **Fixed embeddings.** Learning curves on the MuJoCo benchmark, without fixed embeddings for the value function and policy, so that the current encoder is used rather than the fixed encoder. Results are averaged over 10 seeds. The shaded area captures a 95% confidence interval around the average performance.

### D.3 Normalization

SALE normalizes embeddings and input through our proposed AvgL1Norm:

$$\text{AvgL1Norm}(x) := \frac{x}{\frac{1}{N} \sum_i |x_i|}. \tag{46}$$

AvgL1Norm is embedded into the architecture used by SALE (see Pseudocode 1), applied to the output of the state encoder $f$ to give the state embedding $z^s = \text{AvgL1Norm}(f(s))$. It is also used following a linear layer in the input of both the value function and policy. This makes the input to the value function and policy:

$$Q(z^{sa}, \text{AvgL1Norm}(f(s)), \text{AvgL1Norm}(\phi^{sa})), \tag{47}$$

$$\pi(\text{AvgL1Norm}(f(s)), \text{AvgL1Norm}(\phi^s)), \tag{48}$$

where $\phi^{sa}$ is the output from a linear layer on the state-action input, $\text{Linear}(s,a)$, and $\phi^s$ is the output from a linear layer on the state input, $\text{Linear}(s)$.

In this section, we vary how normalization is used, swapping which components it is used on and what normalization function is used.

(Figure 13) **Normalization 1:** No normalization on $\phi$. Normalization is not applied to either $\phi$. This makes the input to the value function and policy:

$$Q(z^{sa}, \text{AvgL1Norm}(f(s)), \phi^{sa}), \tag{49}$$

$$\pi(\text{AvgL1Norm}(f(s)), \phi^s). \tag{50}$$

(Figure 13) **Normalization 2:** No normalization. Normalization is not applied anywhere. This makes the input to the value function and policy:

$$Q(z^{sa}, f(s), \phi^{sa}), \tag{51}$$

$$\pi(f(s), \phi^s). \tag{52}$$

(Figure 14) **Normalization 3:** Normalization used on $z^{sa}$. Since $z^{sa}$ is trained to predict the next state embedding $z^{s'}$, which is normalized, normalization may not be needed for the output of the state-action encoder $g$. We try also applying normalization to the output of $g$. This makes the input to the value function and policy:

$$Q(\text{AvgL1Norm}(g(z^s, a)), \text{AvgL1Norm}(f(s)), \text{AvgL1Norm}(\phi^{sa})), \tag{53}$$

$$\pi(\text{AvgL1Norm}(f(s)), \text{AvgL1Norm}(\phi^s)). \tag{54}$$

(Figure 15) **Normalization 4:** BatchNorm. BatchNorm is used instead of AvgL1Norm. This makes the input to the value function and policy:

$$Q(z^{sa}, \text{BatchNorm}(f(s)), \text{BatchNorm}(\phi^{sa})), \tag{55}$$

$$\pi(\text{BatchNorm}(f(s)), \text{BatchNorm}(\phi^s)), \tag{56}$$

(Figure 15) **Normalization 5:** LayerNorm. LayerNorm is used instead of AvgL1Norm. This makes the input to the value function and policy:

$$Q(z^{sa}, \text{LayerNorm}(f(s)), \text{LayerNorm}(\phi^{sa})), \tag{57}$$

$$\pi(\text{LayerNorm}(f(s)), \text{LayerNorm}(\phi^s)), \tag{58}$$

(Figure 16) **Normalization 3:** Cosine similarity loss. This is inspired by SPR [Schwarzer et al., 2020], where normalization is not used and a cosine similarity loss is used instead of MSE for updating the encoder. This means the encoder loss function, which is originally:

$$\mathcal{L}(f,g) := \left( g(f(s), a) - |f(s')|_\times \right)^2 = \left( z^{sa} - |z^{s'}|_\times \right)^2, \tag{59}$$

now becomes

$$\mathcal{L}(f,g) := \text{Cosine}\left( z^{sa}, |z^{s'}|_\times \right), \tag{60}$$

$$\text{Cosine}\left( z^{sa}, |z^{s'}|_\times \right) := \left( \frac{z^{sa}}{\|z^{sa}\|_2} \right)^\top \left( \frac{|z^{s'}|_\times}{\||z^{s'}|_\times\|_2} \right). \tag{61}$$

No normalization is used elsewhere.

Table 10: Average performance on the MuJoCo benchmark at 1M time steps. ± captures a 95% confidence interval around the average performance. Results are over 10 seeds.

| Algorithm | HalfCheetah | Hopper | Walker2d | Ant | Humanoid |
|---|---|---|---|---|---|
| TD7 (no checkpoints) | $17123 \pm 296$ | $3361 \pm 429$ | $5718 \pm 308$ | $8605 \pm 1008$ | $7381 \pm 172$ |
| TD3 | $10574 \pm 897$ | $3226 \pm 315$ | $3946 \pm 292$ | $3942 \pm 1030$ | $5165 \pm 145$ |
| No normalization on $\phi$ | $17231 \pm 246$ | $2647 \pm 466$ | $5639 \pm 1248$ | $8191 \pm 846$ | $5862 \pm 1471$ |
| No normalization | $17275 \pm 288$ | $3359 \pm 479$ | $6168 \pm 164$ | $7274 \pm 662$ | $4803 \pm 1706$ |
| Normalization on $z^{sa}$ | $16947 \pm 284$ | $3383 \pm 262$ | $5502 \pm 1142$ | $8049 \pm 538$ | $6418 \pm 302$ |
| BatchNorm | $17299 \pm 218$ | $2318 \pm 473$ | $4839 \pm 987$ | $7636 \pm 769$ | $6979 \pm 325$ |
| LayerNorm | $17132 \pm 360$ | $2451 \pm 902$ | $4470 \pm 1096$ | $6331 \pm 1024$ | $6712 \pm 1297$ |
| Cosine similarity loss | $16897 \pm 372$ | $3324 \pm 249$ | $5566 \pm 353$ | $7873 \pm 511$ | $4370 \pm 1573$ |

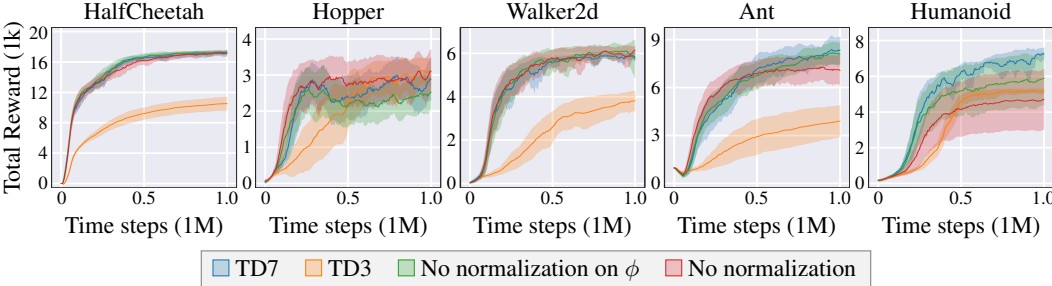

Figure 13: **Removing normalization.** Learning curves on the MuJoCo benchmark, where normalization is removed on certain components of the input to the value function and policy. Results are averaged over 10 seeds. The shaded area captures a 95% confidence interval around the average performance.

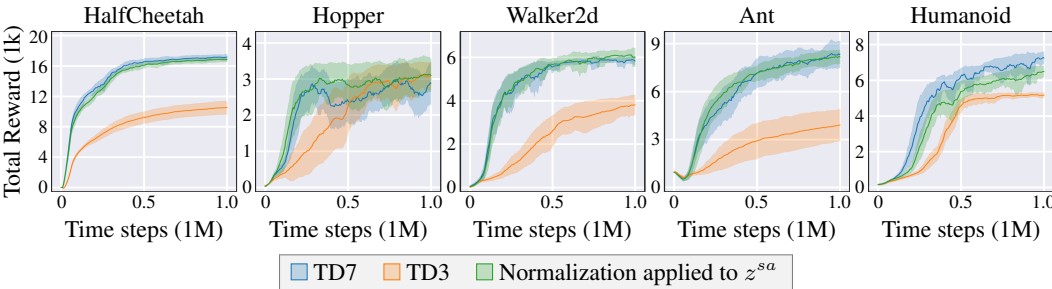

Figure 14: **Normalization on $z^{sa}$.** Learning curves on the MuJoCo benchmark, where normalization is also applied to the state-action embedding $z^{sa}$. Results are averaged over 10 seeds. The shaded area captures a 95% confidence interval around the average performance.

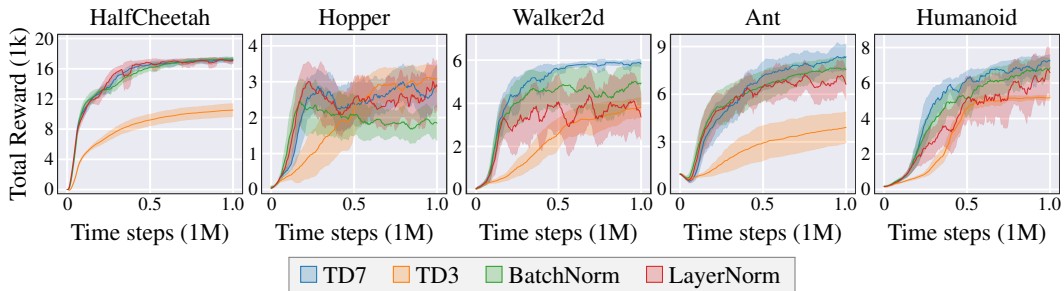

Figure 15: **Alternate normalization.** Learning curves on the MuJoCo benchmark, where BatchNorm or LayerNorm is used instead of AvgL1Norm. Results are averaged over 10 seeds. The shaded area captures a 95% confidence interval around the average performance.

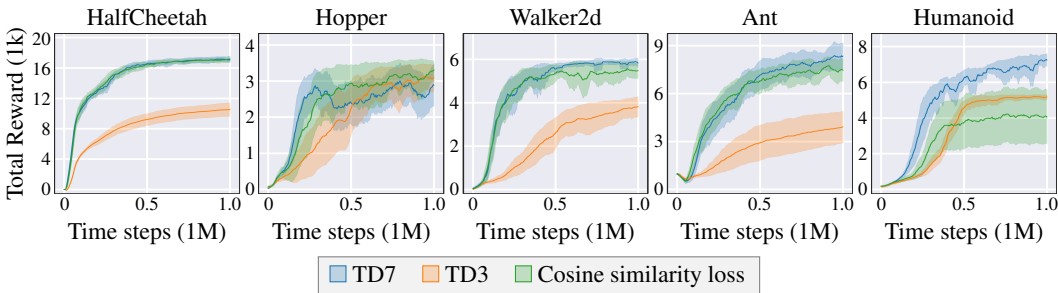

Figure 16: **Cosine similarity instead of normalization.** Learning curves on the MuJoCo benchmark, where normalization is removed and replaced with a cosine similarity loss, as inspired by SPR [Schwarzer et al., 2020]. Results are averaged over 10 seeds. The shaded area captures a 95% confidence interval around the average performance.

## D.4 End-to-end

The embeddings used in SALE are decoupled, meaning that the encoders used to output the embeddings are trained independently from the value function or policy. Instead, we might consider training the encoders end-to-end with the value function. Doing so requires training the current embedding with the value function loss (rather than using the fixed embedding). This means that the encoders are trained with the following loss:

$$\mathcal{L}(f_{t+1}, g_{t+1}) := \text{Huber}\Big(\text{target} - Q_{t+1}(z^{sa}_{t+1}, z^s_{t+1}, s, a)\Big) + \beta \left(z^{sa}_{t+1} - |z^{s'}_{t+1}|_{\times}\right)^2, \quad (62)$$

$$\text{target} := r + \gamma \, \text{clip}\big(\min\left(Q_{t,1}(x), Q_{t,2}(x)\right), Q_{\min}, Q_{\max}\big), \quad (63)$$

$$x := [z^{s'a'}_t, z^{s'}_t, s', a'], \quad (64)$$

$$a' := \pi_t(z^{s'}_t, s') + \epsilon, \quad (65)$$

$$\epsilon \sim \text{clip}(\mathcal{N}(0, \sigma^2), -c, c). \quad (66)$$

In Table 11 and Figure 17 we display the performance when varying the hyperparameter $\beta$.

Table 11: Average performance on the MuJoCo benchmark at 1M time steps. $\pm$ captures a $95\%$ confidence interval around the average performance. Results are over 10 seeds.

| Algorithm | HalfCheetah | Hopper | Walker2d | Ant | Humanoid |
|---|---|---|---|---|---|
| TD7 (no checkpoints) | $17123 \pm 296$ | $3361 \pm 429$ | $5718 \pm 308$ | $8605 \pm 1008$ | $7381 \pm 172$ |
| TD3 | $10574 \pm 897$ | $3226 \pm 315$ | $3946 \pm 292$ | $3942 \pm 1030$ | $5165 \pm 145$ |
| End-to-end, 0.1 | $16186 \pm 360$ | $1820 \pm 674$ | $5013 \pm 729$ | $6601 \pm 1251$ | $5076 \pm 897$ |
| End-to-end, 1 | $15775 \pm 658$ | $1779 \pm 537$ | $4882 \pm 710$ | $6604 \pm 1135$ | $5880 \pm 703$ |
| End-to-end, 10 | $16472 \pm 365$ | $1534 \pm 341$ | $4900 \pm 760$ | $6626 \pm 1253$ | $5279 \pm 1095$ |

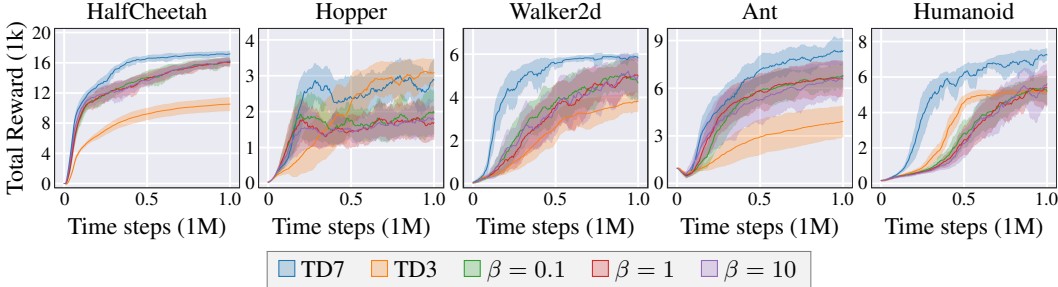

Figure 17: **End-to-end learning.** Learning curves on the MuJoCo benchmark, when training the embeddings end-to-end with the value function, where the encoder loss is weighted with respect to $\beta$. Results are averaged over 10 seeds. The shaded area captures a $95\%$ confidence interval around the average performance.

# E    Extrapolation Error Study

## E.1    Extrapolation Error

Extrapolation error is an issue with deep RL where the value function poorly estimates the value of unseen or rare actions [Fujimoto et al., 2019]. If we assume that $(s, a)$ is not contained in the dataset, then $Q(s, a)$ is effectively a guess generated by the value function $Q$. During training, $s$ is always sampled from the dataset, but $a$ is often generated by the policy ($a \sim \pi(s)$), resulting in a potentially poor estimate.

Extrapolation error is typically considered a problem for offline RL, where the agent is given a fixed dataset and cannot interact further with the environment, as actions sampled from the policy may not be contained in the dataset. Extrapolation error is not considered a problem in online RL since the policy interacts with the environment, collecting data for the corresponding actions it generates.

In our empirical analysis (Section E.2) we observe the presence of extrapolation error when using SALE. Our hypothesis is that by significantly expanding the input dimension dependent on the action, the network becomes more prone to erroneous extrapolations (for example, for the Ant environment, the original action dimension size is $8$, but the state-action embedding $z^{sa}$ has a dimension size of $256$). This is because unseen actions can appear significantly more distinct from seen actions, due to the dramatic increase in the number of dimensions used to represent the action-dependent input.

## E.2    Empirical Analysis

In this section we vary the input to the value function in TD7 to understand the role of the input dimension size and extrapolation error. The default input to the value function in TD7 is as follows:

$$Q(z^{sa}, z^s, \phi), \tag{67}$$
$$\phi := \mathrm{AvgL1Norm}(\mathrm{Linear}(s, a)). \tag{68}$$

Throughout all experiments, we assume no value clipping (as this is introduced in response to the analysis from this section). This makes the value function loss (originally Equation 15):

$$\mathcal{L}(Q_{t+1}) := \mathrm{Huber}\Big(\mathtt{target} - Q_{t+1}(z_t^{sa}, z_t^s, s, a)\Big), \tag{69}$$
$$\mathtt{target} := r + \gamma \min\left(Q_{t,1}(x), Q_{t,2}(x)\right), \tag{70}$$
$$x := [z_{t-1}^{s'a'}, z_{t-1}^{s'}, s', a'], \tag{71}$$
$$a' := \pi_t(z_{t-1}^{s'}, s') + \epsilon, \tag{72}$$
$$\epsilon \sim \mathrm{clip}(\mathcal{N}(0, \sigma^2), -c, c). \tag{73}$$

Policy checkpoints are not used and the policy is not modified. Since extrapolation error is closely linked to available data, we also vary the maximum size of the replay buffer (default: 1M).

(Figure 18) **Extrapolation 1:** No clipping. This is TD7 without value clipping in the value function loss as discussed above.

(Figure 19) **Extrapolation 2:** No $z^{sa}$. We remove $z^{sa}$ from the value function input. This makes the input: $Q_{t+1}(z_t^s, \phi)$.

(Figure 20) **Extrapolation 3:** Small $\phi$. We reduce the number of dimensions of $\phi$ from 256 to 16.

(Figure 21) **Extrapolation 4:** No $z^{sa}$ and small $\phi$. We remove $z^{sa}$ from the value function input and reduce the number of dimensions of $\phi$ from 256 to 16.

(Figure 22) **Extrapolation 5:** Frozen embeddings. The encoders are left unchanged throughout training, by leaving them untrained. This means the input of the value function is $Q_{t+1}(z_0^s, \phi)$.

(Figure 23) **Extrapolation 6:** TD7. The full TD7 method (without checkpoints).

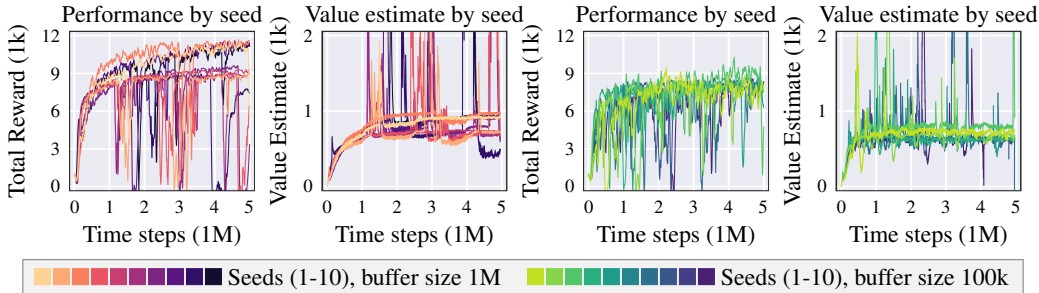

Figure 18: **No clipping.** The performance curve and corresponding value estimate made by the value function, where no value clipping is used. The results for each individual seed are presented for a replay buffer size of either 1M (left) or 100k (right). The environment used is Ant.

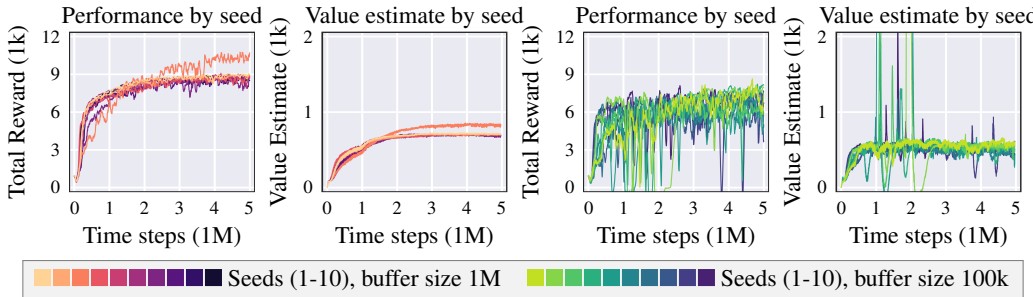

Figure 19: **No $z^{sa}$.** The performance curve and corresponding value estimate made by the value function, where $z^{sa}$ is not included in the input. The results for each individual seed are presented for a replay buffer size of either 1M (left) or 100k (right). The environment used is Ant.

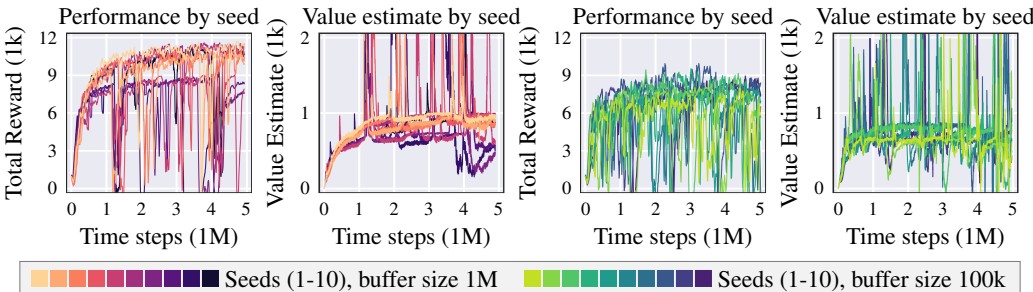

Figure 20: **Small $\phi$.** The performance curve and corresponding value estimate made by the value function, where the dimension size of $\phi$ is reduced. The results for each individual seed are presented for a replay buffer size of either 1M (left) or 100k (right). The environment used is Ant.

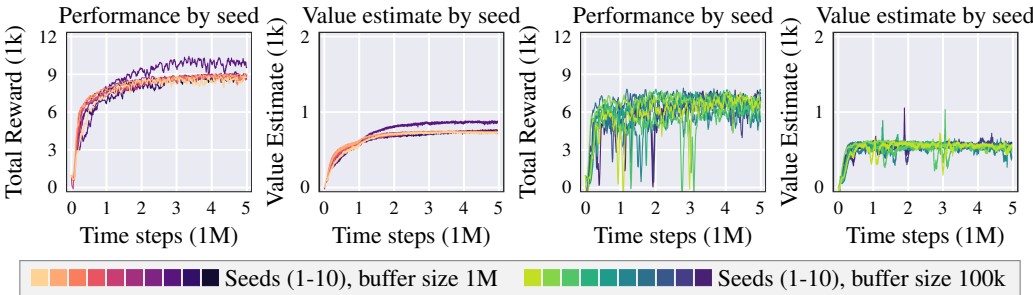

Figure 21: **No** $z^{sa}$**, small** $\phi$**.** The performance curve and corresponding value estimate made by the value function, where $z^{sa}$ is not included in the input and the dimension size of $\phi$ is reduced. The results for each individual seed are presented for a replay buffer size of either 1M (left) or 100k (right). The environment used is Ant.

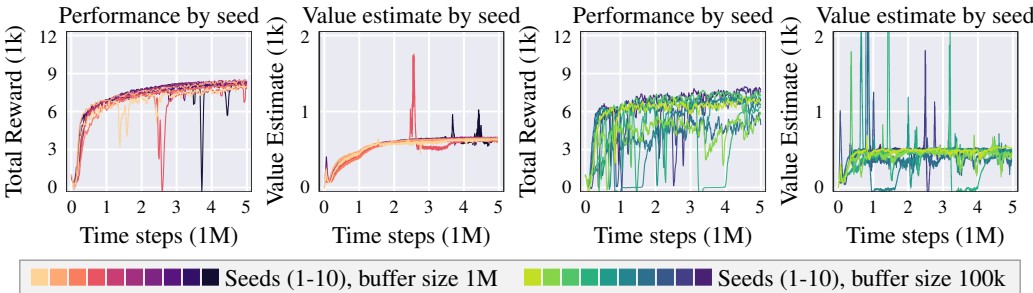

Figure 22: **Frozen embeddings.** The performance curve and corresponding value estimate made by the value function, where the encoders are not trained. The results for each individual seed are presented for a replay buffer size of either 1M (left) or 100k (right). The environment used is Ant.

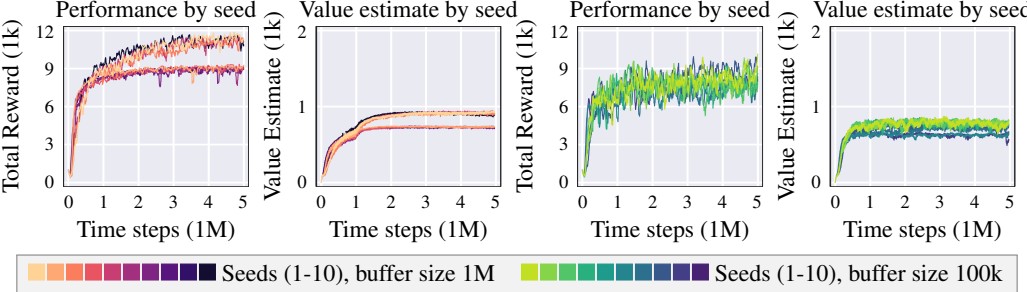

Figure 23: **TD7.** The performance curve and corresponding value estimate made by the value function of TD7 (using value clipping). The results for each individual seed are presented for a replay buffer size of either 1M (left) or 100k (right). The environment used is Ant.

# F    Policy Checkpoints

## F.1    Motivation

Deep RL algorithms are widely known for their inherent instability, which often results in substantial variance in performance during training. Instability in deep RL can occur on a micro timescale (performance can shift dramatically between episodes) [Henderson et al., 2017, Fujimoto and Gu, 2021] and a macro timescale (the algorithm can diverge or collapse with too much training) [Kumar et al., 2020a, Lyle et al., 2021]. In Figure 24 & 25 we show the instability of RL methods by (a) showing how presenting the average performance can hide the instability in a single trial of RL (Figure 24), and (b) measuring how much the performance can drop between nearby evaluations (Figure 25).

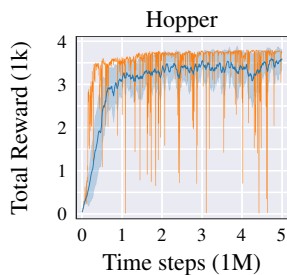

Figure 24: ☐ Standard presentation ■ Single seed
The training curve as commonly presented in RL papers ☐ and the training curve of a single seed ■. Both curves are from the TD3 algorithm, trained for 5M time steps. The standard presentation is to evaluate every $N_{freq}$ steps, average scores over $N_{episodes}$ evaluation episodes and $N_{seeds}$ seeds, then to smooth the curve by averaging over a window of $N_{window}$ evaluations. (In our case this corresponds to $N_{freq} = 5000$, $N_{episodes} = 10$, $N_{seeds} = 10$, $N_{window} = 10$). The learning curve of a single seed has no smoothing over seeds or evaluations ($N_{seeds} = 1$, $N_{window} = 1$). By averaging over many seeds and evaluations, the training curves in RL can appear deceptively smooth and stable.

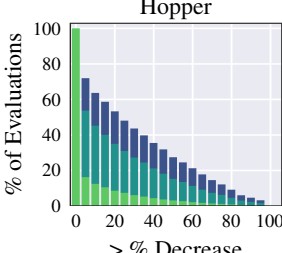

Figure 25: ■ window = 1 ■ window = 5 ■ window = 10
The % of evaluations which suffer from a performance drop of at least $x$% in the next window evaluations for the TD3 algorithm, trained for 5M time steps. For example, for any given evaluation, the likelihood of one of the next 10 evaluations performing at least 50% worse is around 30%. Only evaluations from the last 1M training steps are used, so that early training instability is not accounted for. This shows that large drops in performance are common.

## F.2    Method

The overview for our approach for using checkpoints in RL:

- Assess the current policy (using training episodes).
- Train the current policy (with a number of time steps equal (or proportional) to the number of time steps viewed during assessment).
- If the current policy outperforms the checkpoint policy, then update the checkpoint policy.

Note that we aim to use training episodes to evaluate the checkpoint policy, as any evaluation or test time episodes are considered entirely separate from the training process. For clarity, we will use the term *assessment* (rather than evaluation) to denote any measure of performance which occurs during training. In Algorithm 3 we outline the basic policy checkpoint strategy.

**Minimum performance criteria.** An interesting choice that remains is how to determine whether the current policy outperforms the checkpoint policy. Instead of using the average performance, we consider the minimum performance over a small number of assessment episodes. Using the minimum provides a criteria which is more sensitive to instability, favoring policies which achieve a consistent performance.

Additionally, using the minimum allows us to end the assessment phase early, if the minimum performance of the current policy drops below the minimum performance of the checkpoint policy. This allows us to use a higher maximum number of assessment episodes, while not wasting valuable training episodes assessing suboptimal policies. In Algorithm 4 we outline policy checkpoints where the minimum performance is used, and the assessment phase is variable length due to early termination.

**Algorithm 3** Basic Policy Checkpoints

1: **for** episode $= 1$ **to** assessment_episodes **do**            ▷ *Assessment*
2:      Follow the current policy $\pi_{t+1}$ and determine episode_reward.
3:      Update the performance measure of the current policy.
4:      Increment timesteps_since_training by the length of the episode.

5: **if** actor $\pi_{t+1}$ outperforms checkpoint policy $\pi_c$ **then**       ▷ *Checkpointing*
6:      Update checkpoint networks $\pi_c \leftarrow \pi_{t+1}$, $f_c \leftarrow f_t$.
7:      Update checkpoint performance.

8: **for** $i = 1$ **to** timesteps_since_training **do**            ▷ *Training*
9:      Train RL agent.
10:     Reset the performance measure of the current policy.

---

**Algorithm 4** Policy Checkpoints with Minimum Performance and Early Termination

1: **for** episode $= 1$ **to** assessment_episodes **do**            ▷ *Assessment*
2:      Follow the current policy $\pi_{t+1}$ and determine episode_reward.
3:      min_performance $\leftarrow \min($min_performance, episode_reward$)$.
4:      Increment timesteps_since_training by the length of the episode.

5:      **if** min_performance $\leq$ checkpoint_performance **then**      ▷ *Early termination*
6:         End current assessment.

7: **if** min_performance $\geq$ checkpoint_performance **then**       ▷ *Checkpointing*
8:      Update checkpoint networks $\pi_c \leftarrow \pi_{t+1}$, $f_c \leftarrow f_t$.
9:      checkpoint_performance $\leftarrow$ min_performance

10: **for** $i = 1$ **to** timesteps_since_training **do**          ▷ *Training*
11:     Train RL agent.
12:     Reset min_performance.

---

**Early learning.** In our empirical analysis we observe that long assessment phases had a negative impact on early learning (Table 13). A final adjustment to policy checkpoints is to keep the number of assessment episodes fixed to 1 during the early stages of learning, and then increase it to a much larger value later in learning. Since the policy or environment may be stochastic, when increasing the number of assessment episodes, we reduce the performance of the checkpoint policy (by a factor of 0.9), since the checkpoint performance may be overfit to the single episode used to assess its performance.

In Algorithm 1 in the main text, several variables are referenced. These are as follows:

- checkpoint_condition: The checkpoint condition refers to either (a) the maximum number of assessment episodes being reached (20) or the current minimum performance dropping below the minimum performance of the checkpoint policy.

- outperforms: The maximum number of assessment episodes were reached and the current minimum performance is higher than the minimum performance of the checkpoint policy.

**Variable descriptions.** In Table 3 (in Section A.2), which describes the hyperparameters of TD7, several variables are referenced. These are as follows:

- Checkpoint criteria: the measure used to evaluate the performance of the current and checkpoint policy (default: minimum performance over the assessment episodes).

- Early assessment episodes: the maximum number of assessment episodes during the early stage of learning (default: 1).

- Late assessment episodes: the maximum number of assessment episodes during the late stage of learning (default: 20).

- Early time steps: the duration of the early learning stage in time steps. We consider early learning to be the first 750k time steps.

- Criteria reset weight: The multiplier on the current performance of the checkpoint policy that is applied once, after the early learning stage ends (default: 0.9).

### F.3  Empirical Analysis

TD7 uses a checkpointing scheme which:

- Measures the performance of the current and checkpoint policy by the minimum performance over 20 assessment episodes.
- Uses a variable length assessment phase, where the assessment phase terminates early if the minimum performance of the current policy is less than the minimum performance of the checkpoint policy.
- Uses an early learning stage where the maximum number of assessment episodes starts at 1 and is increased to 20 after 750k time steps.

In this section, we vary some of the choices made when using policy checkpoints and analyze the stability benefits of using checkpoints.

(Figure 26) **Checkpoint 1:** Maximum number of assessment episodes. We vary the maximum number episodes used to assess the performance of the current policy (default 20). Early termination of the assessment phase is still used.

(Figure 27) **Checkpoint 2:** Mean, fixed assessment length. We use the mean performance instead of the minimum (default) to determine if the current policy outperforms the checkpoint policy. In this case, early termination of the assessment phase is not used. The number of assessment episodes is fixed at 20.

(Figure 27) **Checkpoint 3:** Mean, variable assessment length. The mean performance is used to determine if the current policy outperforms the checkpoint policy. Early termination of the assessment phase is used. After each episode, we terminate the assessment phase if the current mean performance is below the checkpoint performance. This means the number of assessment episodes is variable, up to a maximum of 20.

(Figure 28) **Checkpoint 4:** Fixed assessment length. The same number of assessment episodes is used during each assessment phase, without early termination.

(Figure 29) **Checkpoint 5:** Immediate. The number of assessment episodes starts at the corresponding number (as opposed to starting at 1 and increasing to a higher number after the early learning state completes at 750k time steps).

In Table 12 we present the main set of results at 5M time steps. To measure early learning performance, in Table 13 we present the performance of using immediate checkpoints at 300k time steps. To observe stability benefits, in Figure 30 we display the performance of five individual seeds, with and without checkpoints.

Table 12: Average performance on the MuJoCo benchmark at 5M time steps. $\pm$ captures a 95% confidence interval around the average performance. Results are over 10 seeds.

| Algorithm | HalfCheetah | Hopper | Walker2d | Ant | Humanoid |
|---|---|---|---|---|---|
| TD7 | $18165 \pm 255$ | $4075 \pm 225$ | $7397 \pm 454$ | $10133 \pm 966$ | $10281 \pm 588$ |
| TD7 (no checkpoints) | $18328 \pm 331$ | $3851 \pm 372$ | $6519 \pm 209$ | $10388 \pm 1024$ | $9521 \pm 820$ |
| Max 10 episodes | $18257 \pm 338$ | $4208 \pm 52$ | $6856 \pm 273$ | $9890 \pm 661$ | $10689 \pm 576$ |
| Max 50 episodes | $17875 \pm 192$ | $4110 \pm 83$ | $7104 \pm 559$ | $8226 \pm 743$ | $10239 \pm 273$ |
| Mean, fixed | $18137 \pm 417$ | $4061 \pm 160$ | $7347 \pm 459$ | $9866 \pm 804$ | $9850 \pm 497$ |
| Mean, variable | $17943 \pm 291$ | $4217 \pm 9$ | $6845 \pm 476$ | $9614 \pm 849$ | $10784 \pm 188$ |
| Immediate, 10 episodes | $17767 \pm 374$ | $4218 \pm 17$ | $7416 \pm 451$ | $10090 \pm 961$ | $9573 \pm 255$ |
| Immediate, 20 episodes | $17900 \pm 127$ | $4078 \pm 53$ | $7696 \pm 616$ | $9576 \pm 935$ | $10219 \pm 551$ |
| Immediate, 50 episodes | $17619 \pm 275$ | $4033 \pm 154$ | $6712 \pm 313$ | $9716 \pm 1091$ | $9687 \pm 584$ |
| Fixed, 10 episodes | $18209 \pm 332$ | $4130 \pm 56$ | $6890 \pm 410$ | $10357 \pm 893$ | $9682 \pm 1408$ |
| Fixed, 20 episodes | $17925 \pm 334$ | $3777 \pm 337$ | $6882 \pm 452$ | $9716 \pm 772$ | $10471 \pm 560$ |
| Fixed, 50 episodes | $17807 \pm 183$ | $3759 \pm 142$ | $6800 \pm 422$ | $8881 \pm 408$ | $8830 \pm 655$ |

Table 13: Average performance on the MuJoCo benchmark at 300k time steps. $\pm$ captures a 95% confidence interval around the average performance. Results are over 10 seeds.

| Algorithm | HalfCheetah | Hopper | Walker2d | Ant | Humanoid |
|---|---|---|---|---|---|
| TD7 | $15031 \pm 401$ | $2948 \pm 464$ | $5379 \pm 328$ | $6171 \pm 831$ | $5332 \pm 714$ |
| Immediate, 10 episodes | $13778 \pm 856$ | $3306 \pm 34$ | $4595 \pm 343$ | $5977 \pm 764$ | $4660 \pm 735$ |
| Immediate, 20 episodes | $12289 \pm 507$ | $3052 \pm 195$ | $4518 \pm 321$ | $5504 \pm 889$ | $4531 \pm 738$ |
| Immediate, 50 episodes | $6658 \pm 1309$ | $2668 \pm 247$ | $2897 \pm 816$ | $3709 \pm 1848$ | $2060 \pm 620$ |

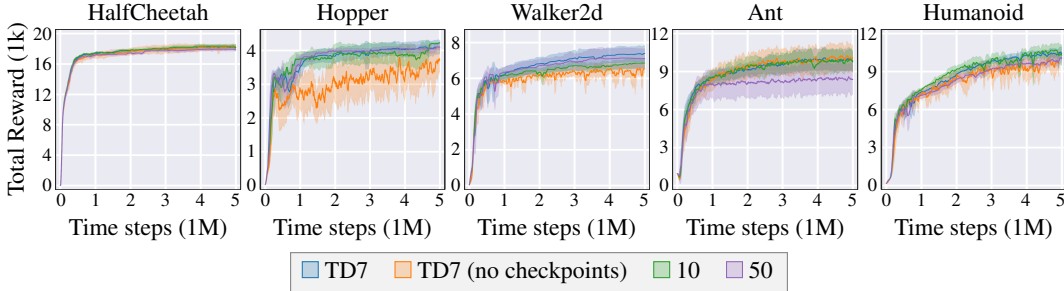

Figure 26: **Maximum number of assessment episodes.** Learning curves on the MuJoCo benchmark, varying the maximum number episodes that the policy is fixed for. Results are averaged over 10 seeds. The shaded area captures a 95% confidence interval around the average performance.

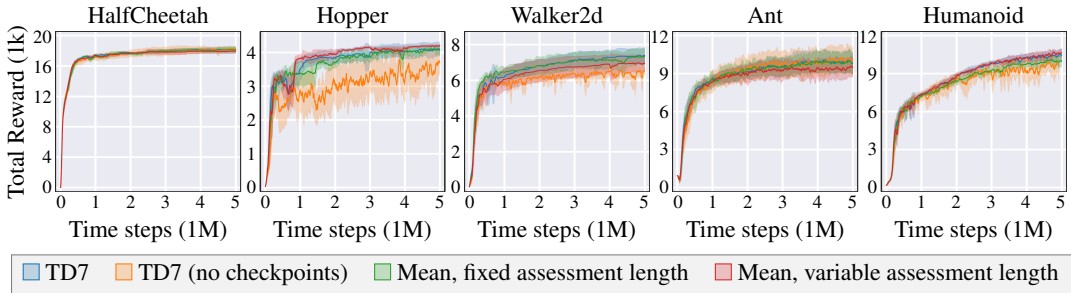

Figure 27: **Checkpoint condition.** Learning curves on the MuJoCo benchmark, using the mean performance of assessment episodes rather than the minimum. The number of assessment episodes is kept at 20. Results are averaged over 10 seeds. The shaded area captures a 95% confidence interval around the average performance.

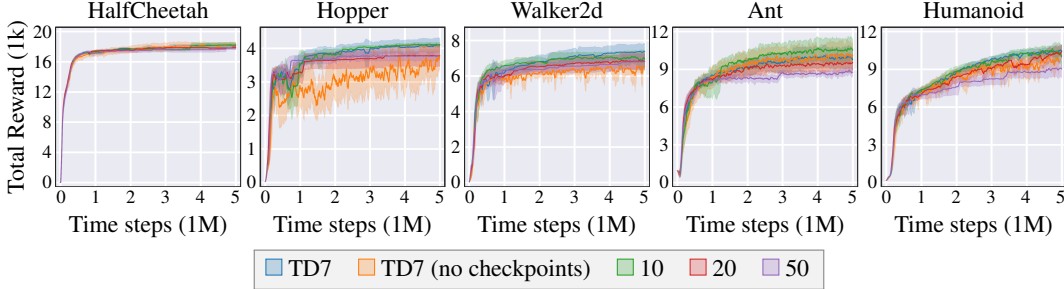

Figure 28: **Fixed assessment length.** Learning curves on the MuJoCo benchmark, where there is no early termination of the assessment phase (i.e the number of assessment episodes is fixed), and the number of assessment episodes is varied. Results are averaged over 10 seeds. The shaded area captures a 95% confidence interval around the average performance.

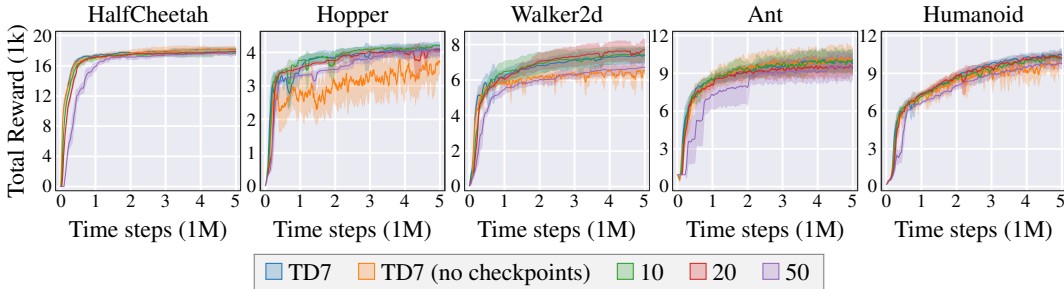

Figure 29: **Immediate.** Learning curves on the MuJoCo benchmark, where the maximum number of assessment episodes does not start at 1 (i.e. there is no early learning phase) and is varied. Results are averaged over 10 seeds. The shaded area captures a 95% confidence interval around the average performance.

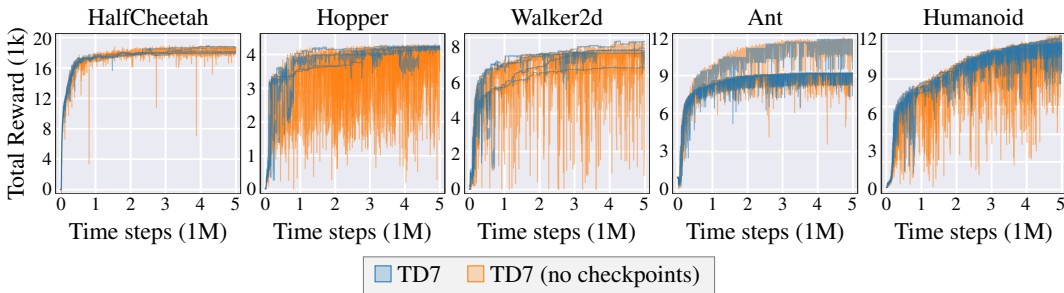

Figure 30: **Performance of individual seeds with and without checkpoints.** Learning curves of five individual seeds, with and without checkpoints, on the MuJoCo benchmark. The shaded area captures a 95% confidence interval around the average performance.

# G  Ablation Study

In this section we perform an ablation study on the components of TD7.

## G.1  Main Components

([Figure 31](#)) **Ablation 1:** No SALE. SALE is entirely removed from TD7. This means the input to the value function and policy return to the original $Q(s, a)$ and $\pi(s)$, respectively. TD7 keeps LAP, policy checkpoints, and any implementation differences over TD3 (discussed in [Section A.1](#)).

([Figure 31](#)) **Ablation 2:** No checkpoints. Similar to TD3 and other off-policy deep RL algorithms, TD7 is trained at every time step, rather than after a batch of episodes. The current policy is used at test time.

([Figure 31](#)) **Ablation 3:** No LAP. TD7 uses the standard replay buffer where transitions are sampled with uniform probability. The value function loss ([Equation 15](#)) uses the mean-squared error (MSE) rather than the Huber loss.

Table 14: Average performance on the MuJoCo benchmark at 5M time steps. $\pm$ captures a 95% confidence interval around the average performance. Results are over 10 seeds.

| Algorithm | HalfCheetah | Hopper | Walker2d | Ant | Humanoid |
|---|---|---|---|---|---|
| TD7 | $18165 \pm 255$ | $4075 \pm 225$ | $7397 \pm 454$ | $10133 \pm 966$ | $10281 \pm 588$ |
| TD3 | $14337 \pm 1491$ | $3682 \pm 83$ | $5078 \pm 343$ | $5589 \pm 758$ | $5433 \pm 245$ |
| No SALE | $17099 \pm 335$ | $4018 \pm 170$ | $6418 \pm 261$ | $7861 \pm 253$ | $7275 \pm 608$ |
| No checkpoints | $18328 \pm 331$ | $3851 \pm 372$ | $6519 \pm 209$ | $10388 \pm 1024$ | $9521 \pm 820$ |
| No LAP | $18104 \pm 315$ | $4188 \pm 22$ | $7233 \pm 251$ | $6940 \pm 1044$ | $10155 \pm 522$ |

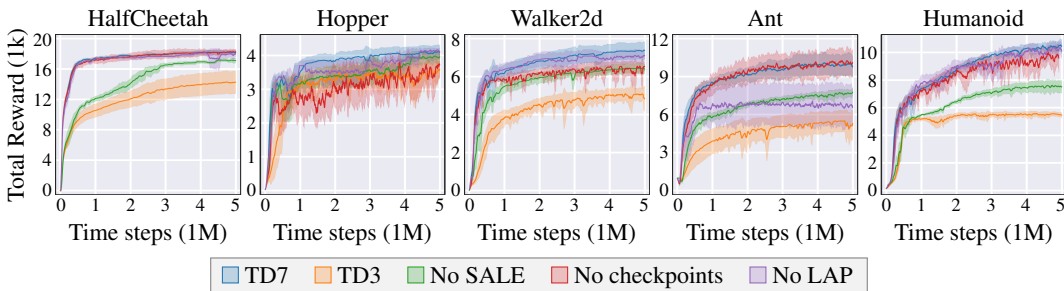

Figure 31: **Main ablation.** Learning curves on the MuJoCo benchmark, removing the main components of TD7. Results are averaged over 10 seeds. The shaded area captures a 95% confidence interval around the average performance.

## G.2 Subcomponents

(Figure 31) **Ablation 4:** No SALE, with encoder. The encoder is kept but trained end-to-end with the value function, rather than via the SALE loss function. This means the encoder can be simply treated as an extension of the architecture of the value function. Since the embeddings trained with the value function may be conflicting with the policy, the policy does not use the state embedding, and only takes its original input $\pi(s)$.

(Figure 31) **Ablation 5:** TD3 with encoder. We add the encoder from SALE to TD3 but train it end-to-end with the value function. The encoder is not applied to the policy.

(Figure 33) **Ablation 6:** Current policy. TD7 is trained in an identical fashion, but uses the current policy at test time, rather than the checkpoint policy.

(Figure 33) **Ablation 7:** TD3 with checkpoints. We add policy checkpoints to TD3, using the checkpoint policy at test time.

(Figure 34) **Ablation 8:** TD3 with LAP. We add LAP to TD3 (identical to the TD3 + LAP [Fujimoto et al., 2020]).

(Figure 35) **Ablation 9:** No clipping. We remove our proposed value clipping for mitigating extrapolation error. This makes the value function loss (originally Equation 15):

$$\mathcal{L}(Q_{t+1}) := \text{Huber}\Big(\texttt{target} - Q_{t+1}(z_t^{sa}, z_t^s, s, a)\Big), \tag{74}$$

$$\texttt{target} := r + \gamma \min \left( Q_{t,1}(x), Q_{t,2}(x) \right), \tag{75}$$

$$x := [z_{t-1}^{s'a'}, z_{t-1}^{s'}, s', a'], \tag{76}$$

$$a' := \pi_t(z_{t-1}^{s'}, s') + \epsilon, \tag{77}$$

$$\epsilon \sim \text{clip}(\mathcal{N}(0, \sigma^2), -c, c). \tag{78}$$

(Figure 35) **Ablation 10:** TD3 with clipping. We add value clipping to the loss function of TD3.

(Figure 36) **Ablation 11:** No normalization. We remove the use of AvgL1Norm from TD7. This is identical to no normalization in the design study in Section D.3, except this version includes policy checkpoints (the design study does not use checkpoints).

(Figure 36) **Ablation 12:** No fixed encoder. We remove the fixed embeddings from the value function and the policy, this means the networks use embeddings from the current encoder, rather than the fixed encoder from the previous iteration. This makes the input: $Q_{t+1}(z_{t+1}^{sa}, z_{t+1}^s, s, a)$ and $\pi_{t+1}(z_{t+1}^s, s)$. This is identical to no normalization in the design study in Section D.3, except this version includes policy checkpoints (the design study does not use checkpoints).

(Figure 37) **Ablation 13:** No implementation. We remove the implementation details differences discussed in Section A.1, namely using both value functions when updating the policy, ELU activation functions in the value function network. The target network update is not changed as this may influence the use of fixed encoders.

(Figure 37) **Ablation 14:** Our TD3. TD3 with the implementation detail differences discussed in Section A.1, namely using both value functions when updating the policy, ELU activation functions in the value function network, and updating the target network every 250 time steps, rather than using an exponential moving average.

Table 15: Average performance on the MuJoCo benchmark at 1M time steps. $\pm$ captures a 95% confidence interval around the average performance. Results are over 10 seeds.

| Algorithm | HalfCheetah | Hopper | Walker2d | Ant | Humanoid |
|---|---|---|---|---|---|
| TD7 | $17434 \pm 155$ | $3512 \pm 315$ | $6097 \pm 570$ | $8509 \pm 422$ | $7429 \pm 153$ |
| TD3 | $10574 \pm 897$ | $3226 \pm 315$ | $3946 \pm 292$ | $3942 \pm 1030$ | $5165 \pm 145$ |
| No SALE | $12981 \pm 261$ | $3536 \pm 65$ | $5237 \pm 376$ | $5296 \pm 1336$ | $6263 \pm 289$ |
| No SALE, with encoder | $15639 \pm 548$ | $3544 \pm 371$ | $5350 \pm 427$ | $6003 \pm 920$ | $5480 \pm 83$ |
| TD3 with encoder | $12495 \pm 813$ | $1750 \pm 302$ | $4226 \pm 491$ | $5255 \pm 579$ | $5082 \pm 317$ |
| No checkpoints | $17123 \pm 296$ | $3361 \pm 429$ | $5718 \pm 308$ | $8605 \pm 1008$ | $7381 \pm 172$ |
| Current policy | $17420 \pm 273$ | $2940 \pm 636$ | $5765 \pm 800$ | $8748 \pm 397$ | $7162 \pm 274$ |
| TD3 with checkpoints | $10255 \pm 656$ | $3414 \pm 77$ | $3266 \pm 474$ | $3843 \pm 749$ | $5349 \pm 72$ |
| No LAP | $17347 \pm 207$ | $3697 \pm 144$ | $6382 \pm 339$ | $6571 \pm 1504$ | $8082 \pm 260$ |
| TD3 with LAP | $10324 \pm 1159$ | $3117 \pm 554$ | $4127 \pm 330$ | $4310 \pm 1150$ | $5090 \pm 190$ |
| No clipping | $17378 \pm 100$ | $3762 \pm 118$ | $6198 \pm 289$ | $7695 \pm 497$ | $7251 \pm 274$ |
| TD3 with clipping | $10283 \pm 422$ | $2969 \pm 682$ | $3990 \pm 258$ | $3711 \pm 799$ | $5254 \pm 203$ |
| No normalization | $17391 \pm 275$ | $3640 \pm 95$ | $6256 \pm 317$ | $7807 \pm 266$ | $4829 \pm 1809$ |
| No fixed encoder | $17145 \pm 138$ | $3710 \pm 120$ | $5869 \pm 531$ | $8287 \pm 379$ | $7412 \pm 337$ |
| No implementation | $17334 \pm 99$ | $3750 \pm 136$ | $5819 \pm 71$ | $7756 \pm 704$ | $7042 \pm 259$ |
| Our TD3 | $11068 \pm 1399$ | $2791 \pm 632$ | $4179 \pm 297$ | $5489 \pm 448$ | $5186 \pm 108$ |

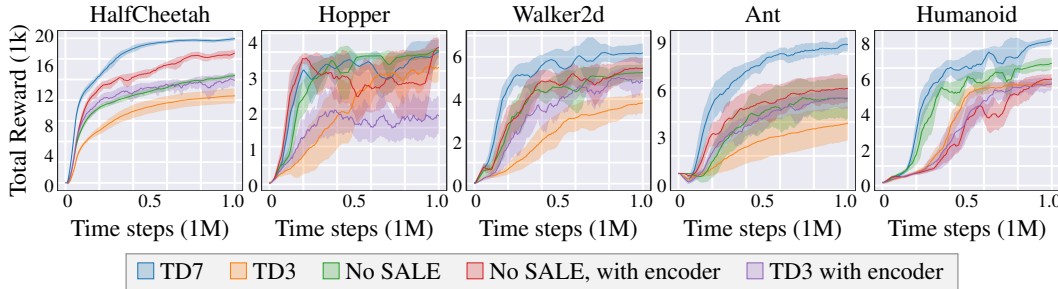

Figure 32: **SALE.** Learning curves on the MuJoCo benchmark, varying the usage of SALE. Results are averaged over 10 seeds. The shaded area captures a 95% confidence interval around the average performance.

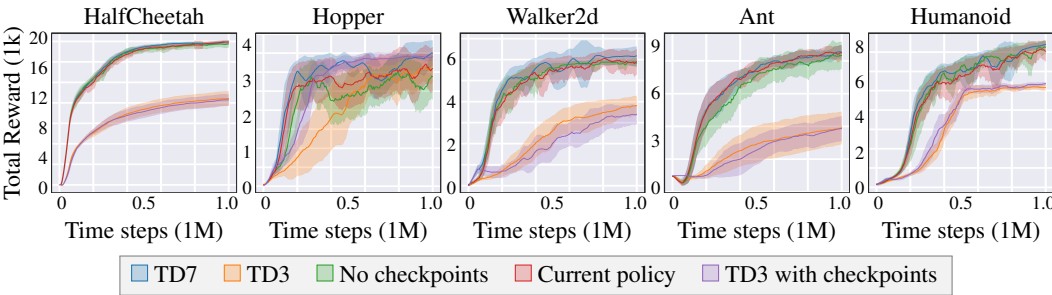

Figure 33: **Checkpoints.** Learning curves on the MuJoCo benchmark, varying the usage of policy checkpoints. Results are averaged over 10 seeds. The shaded area captures a 95% confidence interval around the average performance.

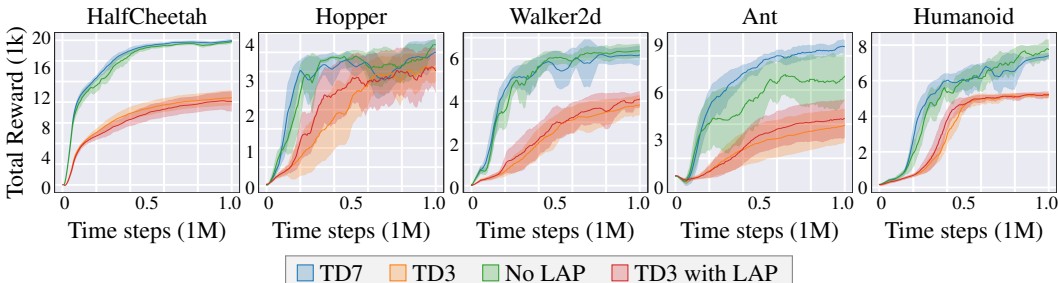

Figure 34: **LAP.** Learning curves on the MuJoCo benchmark, varying the usage of LAP. Results are averaged over 10 seeds. The shaded area captures a 95% confidence interval around the average performance.

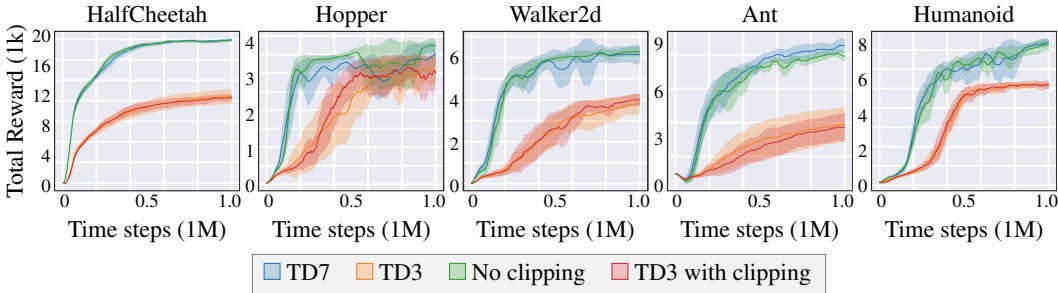

Figure 35: **Clipping.** Learning curves on the MuJoCo benchmark, varying the usage of our proposed value clipping for mitigating extrapolation error. Results are averaged over 10 seeds. The shaded area captures a 95% confidence interval around the average performance.

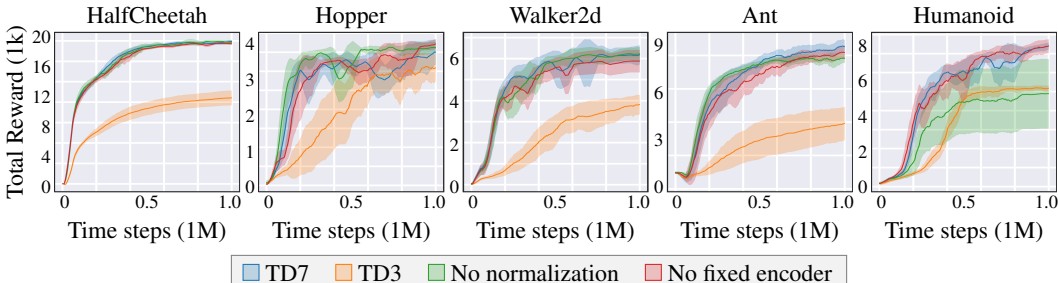

Figure 36: **SALE components.** Learning curves on the MuJoCo benchmark, removing components of SALE. Results are averaged over 10 seeds. The shaded area captures a 95% confidence interval around the average performance.

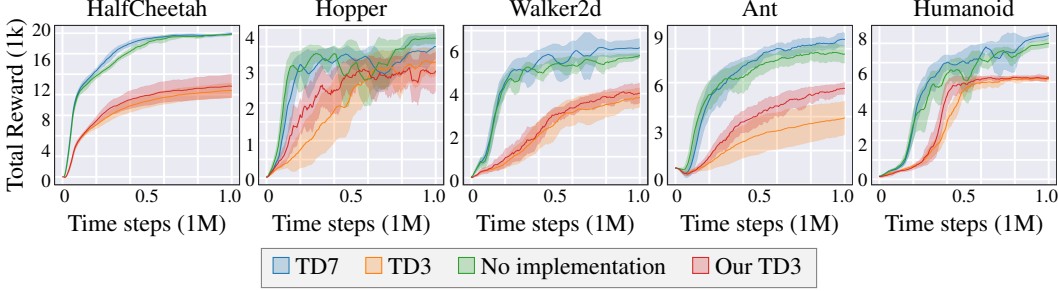

Figure 37: **Implementation differences.** Learning curves on the MuJoCo benchmark, varying the minor implementation details between TD3 and TD7. Results are averaged over 10 seeds. The shaded area captures a 95% confidence interval around the average performance.

# H   Offline RL Learning Curves

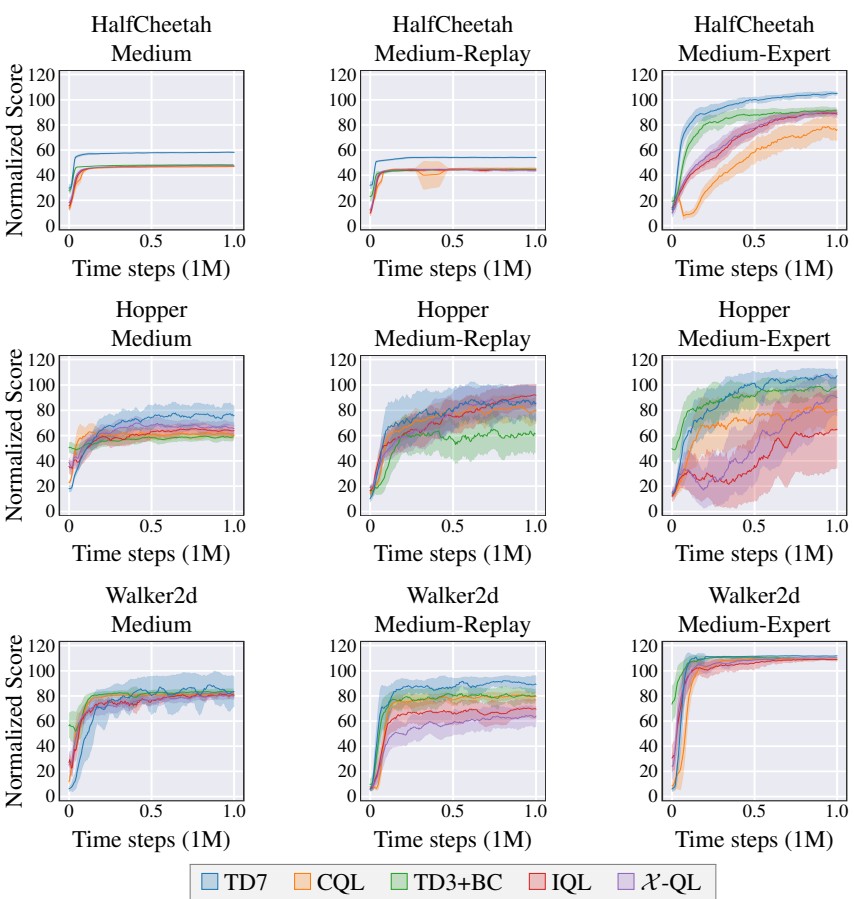

Figure 38: Learning curves on the offline D4RL benchmark. Results are averaged over 10 seeds. The shaded area captures a 95% confidence interval around the average performance.

# I   Run time

We benchmark the run time of TD7 and the baseline algorithms. Evaluation episodes were not included in this analysis. Each algorithm is trained using the same deep learning framework, PyTorch [Paszke et al., 2019]. All experiments are run on a single Nvidia Titan X GPU and Intel Core i7-7700k CPU. Software is detailed in Appendix B. We display run time adjusted learning curves in Figure 39.

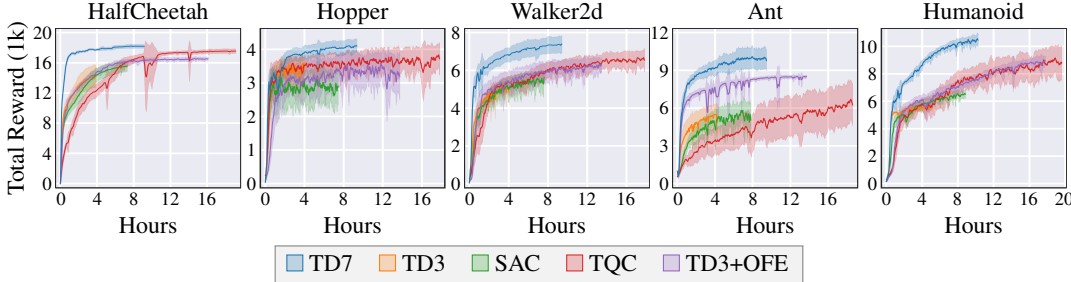

Figure 39: **Run time adjusted curves.** Learning curves on the MuJoCo benchmark over 5M time steps, where the x-axis is run time. Results are averaged over 10 seeds. The shaded area captures a 95% confidence interval around the average performance.

## J  Limitations

**Depth vs. Breadth.** Although our empirical study is extensive, the scope of our experimentation emphasizes depth over breadth. There remains many other interesting benchmarks for continuous control, such as examining a setting with image-based observations.

**Baselines.** We benchmark TD7 against the highest performing agents in both the online and offline setting. However, due to the fact that most representation learning methods do not cover low level states, we only benchmark against a single other method focused on representation learning. While it would be surprising if a method, originally designed for a different setting, outperformed the strongest existing baselines, exploring additional methods could provide new insight into representation learning for low level states.

**Computational cost.** While TD7 has a lower computational cost than other competing methods, the run time over TD3 (the base algorithm) is more than double.

**Theoretical results.** We perform extensive empirical analysis to uncover which factors of dynamics-based representation learning are important for performance. However, we do not address the theoretical side of representation learning. There are many important questions regarding why dynamics-based representation learning is effective, or what makes a good representation.

