# OpenReview forum: "For SALE: State-Action Representation Learning for Deep Reinforcement Learning"
_NeurIPS.cc/2023/Conference — NeurIPS 2023 poster_

### Official Review · Reviewer_FeJr · 2023-07-03

**Soundness:** 3 good
**Presentation:** 3 good
**Contribution:** 3 good
**Rating:** 6
**Confidence:** 4

**Summary:**

This paper proposes a new RL algorithm called TD7, which is based on TD3, and adopts additional techniques including (1) learning state-action representations, (2) LAP prioritized replay, (3) a behavior cloning term in the learning objective, and (4) checkpoints, where (3) and (4) are applied exclusively to the offline RL setting. They demonstrate the superior empirical performance of TD7 on the D4RL benchmark.

**Strengths:**

1. The paper is well written and easy to follow.
2. Extensive experiments and ablation studies to evaluate the proposed methods.
3. The proposed method is shown to have strong empirical performance on established benchmarks.

**Weaknesses:**

1. **Limited contribution**: While I appreciate TD7 as a competent algorithm for both online and offline RL, I'm not so sure about the contribution of SALE especially regarding its applicability to other existing baselines. It doesn't seem conclusive to me that the proposed embedding is generally superior for RL agents. From the ablation studies it seems many independent design choices could significantly affect the final performance, which may also be algorithm dependent.

**Questions:**

1. Could you provide more evidence that shows SALE's applicability to other existing RL methods?

**Limitations:**

The limitations are not explicitly discussed in the paper.

---

> ### Author Rebuttal · Authors · 2023-08-08
>
> Thank you for the review and the feedback.
>
> **Combining SALE with other algorithms:** Thank you for bringing up this concern. In the table below we have included results with SALE applied to SAC.  Learning curves are included in the PDF of the general response (Figure 2). No modifications or hyperparameter changes were made to either SALE or SAC, we simply applied SALE directly. We hope this addresses your primary concern!
>
> |             | SAC + SALE          | SAC                |
> |-------------|---------------------|--------------------|
> | HalfCheetah | **17316 $\pm$ 236** | 15526 $\pm$ 697    |
> | Hopper      | 2842 $\pm$ 951      | **3167 $\pm$ 485** |
> | Walker2d    | **6887 $\pm$ 306**  | 5681 $\pm$ 329     |
> | Ant         | **9288 $\pm$ 462**  | 4615 $\pm$ 2022    |
> | Humanoid    | **9062 $\pm$ 335**  | 6555 $\pm$ 279     |
>
> Results are for 5M time steps, and 10 seeds. $\pm$ captures a 95% confidence interval.
>
> This is further evidence that while there are many design choices to be made, we are presenting a set of design choices which are robust to algorithm and task. Regardless, it is our hope that by presenting an in-depth empirical study, readers can determine which choices are more important than others, and can start with a good default set.
>
> **Limitations:** As stated in the checklist, these are included in Appendix J. Sorry for the misunderstanding, space was a concern in the paper.

---

> > ### Comment · Reviewer_FeJr · 2023-08-16
> >
> > Thank you for the response. I have increased my rating as my main concern on the compatibility with existing baselines has been addressed.

---

### Official Review · Reviewer_xwBa · 2023-07-05

**Soundness:** 3 good
**Presentation:** 3 good
**Contribution:** 3 good
**Rating:** 7
**Confidence:** 4

**Summary:**

The paper introduces an approach dubbed SALE for learning embeddings that model the interaction between state and action in low-level state environments. The authors extensively study the design space of these embeddings and integrate SALE into the TD3 algorithm along with 3 other components to form a new algorithm, TD7.
They perform an extensive empirical evaluation over the design space to discover the most significant contributors to final performance. The paper shows that TD7 outperforms existing continuous control algorithms on MuJoCo OpenAI gym tasks.

**Strengths:**

**Originality**
The paper's originality lies in its comprehensive study of learning embeddings that model the interaction between state and action in low-level state environments. The experiments conducted provide evidence of the effectiveness of their approach in improving the performance of TD3.

**Quality**
See Weaknesses Section.

**Clarity**
The paper is well-written and organized, making it easy for readers to follow the authors' thought process and understand the methodology and results.

**Significance**
The authors demonstrate that their proposed method, when integrated into TD3 to form what they name TD7, performs significantly better than TD3 for the MuJoCo environments. This work brings to light the importance of learning the interaction between state and action information to improve performance in RL.

**Weaknesses:**

While the paper presents a novel approach to learning embeddings for state and action information for RL, there are areas where it could be improved.

Firstly, the evaluation of TD7 could be expanded to include more benchmarks. While the MuJoCo environments provide valuable data, including additional benchmarks such as Procgen or Brax could offer a more comprehensive understanding of the method's performance and any potential limitations.

These considerations, while not detracting from the originality and potential significance of the method, do highlight areas where further work could be beneficial.

**Questions:**

See Weaknesses Section.

**Limitations:**

The authors adequately addressed the limitations.

---

> ### Author Rebuttal · Authors · 2023-08-02
>
> **Is part of this review missing?**
>
> We just wanted to double check that this review is complete. It currently reads as if some of the weaknesses were mistakenly deleted. If some other weaknesses did indeed get removed, let us know so we can improve the paper accordingly. Regardless, thank you for your time and the review. We address your comments below.
>
> **TD7 on other benchmarks:** To address this concern, we have collected results for TD7 on the DMC suite. As shown in the table below, TD7 consistently maintains its advantage over TD3 in DMC as well. In the attached PDF of the general response, we include the full learning curves for these environments (Figure 1). We hope this addresses your primary concern!
>
> |                         | TD7                 | TD3                |
> |-------------------------|---------------------|--------------------|
> | Ball in Cup Catch       | **982 $\pm$ 1.8**   | 981 $\pm$ 3.3      |
> | Cartpole Balance Sparse | **1000 $\pm$ 0.0**  | **1000 $\pm$ 0.0** |
> | Cartpole Swingup        | **875 $\pm$ 5.8**   | 871 $\pm$ 6.6      |
> | Cheetah Run             | **914 $\pm$ 2.9**   | 866 $\pm$ 23.5     |
> | Finger Spin             | **989 $\pm$ 1.4**   | 975 $\pm$ 14.1     |
> | Finger Turn Hard        | **742 $\pm$ 136.8** | 552 $\pm$ 80.7     |
> | Hopper Stand            | **774 $\pm$ 177.4** | 480 $\pm$ 180.8    |
> | Hopper Hop              | **175 $\pm$ 98.7**  | 58 $\pm$ 42.3      |
> | Humanoid Stand          | **504 $\pm$ 336.5** | 7 $\pm$ 0.4        |
> | Humanoid Walk           | **389 $\pm$ 157.6** | 36 $\pm$ 72.4      |
> | Walker Stand            | **991 $\pm$ 1.5**   | 985 $\pm$ 3.1      |
> | Walker Walk             | **981 $\pm$ 2.7**   | 965 $\pm$ 5.2      |
> | Walker Run              | **807 $\pm$ 62.5**  | 631 $\pm$ 74.7     |
> | Quadruped Walk          | **950 $\pm$ 4.9**   | 750 $\pm$ 168.9    |
> | Quadruped Run           | **684 $\pm$ 87.5**  | 503 $\pm$ 193.7    |
> | Mean                    | **784**             | 644                |
> | Median                  | **875**             | 750                |
> | IQM                     | **856**             | 733                |
>
> Results are for 1M time steps and 10 seeds. $\pm$ captures a 95% confidence interval.

---

> > ### Comment · Reviewer_xwBa · 2023-08-19
> >
> > The results for TD7 in comparison to TD3 are compelling and offer a clearer picture of the algorithm's performance across a diverse set of tasks.
> > Given the updated information and the detailed results you've shared, I am pleased to increase my score.

---

### Official Review · Reviewer_YfJA · 2023-07-06

**Soundness:** 4 excellent
**Presentation:** 3 good
**Contribution:** 3 good
**Rating:** 7
**Confidence:** 3

**Summary:**

This work introduces a novel state-action representation learning framework SALE and two other techniques (e.g. checkpointing, a new type of Q value clipping) that substantially improve the data efficiency and final performance of TD3 in online and offline RL.

**Strengths:**

1. The work studies joint state-action representation learning, which is under-studied with respect to state representation learning.
2. The method is very clear, and the authors provide extensive empirical analysis and ablation explaining their design choices.
3. Some techniques introduced in this work (checkpointing, Q clipping in Eq. 6) can be applied to most RL methods and are thus influential beyond representation learning.


**Weaknesses:**

**Core comments:**
1. I believe the paper would read more easily after reorganization.
    * Section 4.2 feels like an ablation study that should follow the core empirical results. Also, Section 4.2 references TD7 results, though TD7 is not introduced until section 5.
    * Section 5 begins by mentioning stability and extrapolation error jumped out at me; the intro and related work emphasize the representation learning side of this work much more, and extrapolation error is not mentioned in sections 1-4. I believe the paper would read better if the authors painted a clearer picture of what the stability side of the work entails in the introduction, and included transitions between section 4 and 5 that clearly indicate “we talked about one issue, and now we’re going to talk about another.”
    * Transition sentences before bolded subheadings (e.g. Normalizing embeddings, Fixed embeddings) would help in Section 4 – something to the effect of “We now discuss two important aspect of SALE…”
1. The checkpoint policy $\pi_c$ denotes the policy is the largest minimum return during the assessment phase. During agent evaluation, you compute the return over N rollouts of $\pi_c$ rather than $\pi_t$, correct? If so, I’m a bit skeptical of the results; is it possible that the perceived benefits of checkpointing are simply due to the batched nature of training it requires? Suppose you run vanilla TD3 within the checkpointing framework but perform separate evaluations using the checkpoint policy *and* current policy. This experiment would be equivalent to vanilla TD3 with variable length batched updates. Does the batched updates alone improve performance?


**Minor comments:**
1. Figure 2 would be more readable with error bars rather than text to indicate the 95% confidence interval.

**Questions:**

1. In principle, any latent state/action RL framework can be modified such that  $Q$ and $\pi$ take both the original state/action and the latent state/action as inputs. Would it be fair to say the ablations in Fig. 2 suggests that any state/action representation learning method should have $Q$ and $\pi$ depend on both the original and latent state/actions?
1. Fig. 2: Can the authors clarify what is meant by percent loss here? Are these quantities all with respect to default values? Where do the default values fall on these figures?
1. The authors mention that checkpointing harms performance early in training – is this because the policy changes rapidly early on? Can the authors provide more detail?
1. Line 217: How does checkpointing change the relevance of data?

**Limitations:**

See Weaknesses.

---

> ### Author Rebuttal · Authors · 2023-08-08
>
> Thank you for the review and helpful comments.
>
> **Reorganization and writing:** Thank you for bringing this up and providing concrete suggestions. We will add transition statements and expand the introduction to clearly establish the challenges associated with our section on stability. We will also look at moving section 4.2 to a later part of the paper, although it is harder to promise this change without first doing so and making sure the flow of the paper still makes sense.
>
> **Does the batched updates alone improve performance?** Great question. We show this result for TD7 in Figure 25 and Table 13 in Appendix G, where “current policy” refers to TD7 trained in an identical fashion but we evaluate with the current policy rather than the checkpoint policy. The result only goes to 1M time steps however, so in the table below we present results for the full 5M time steps.
>
> |             | TD7 (checkpoints, checkpoint policy) | TD7 (checkpoints, current policy) | TD7 (no checkpoints) |
> |-------------|--------------------------------------|-----------------------------------|----------------------|
> | HalfCheetah | 18165 $\pm$ 255                      | 18214 $\pm$ 237                   | **18328 $\pm$ 331**  |
> | Hopper      | **4075 $\pm$ 225**                   | 3690 $\pm$ 562                    | 3851 $\pm$ 372       |
> | Walker2d    | **7397 $\pm$ 454**                   | 7258 $\pm$ 469                    | 6519 $\pm$ 209       |
> | Ant         | 10133 $\pm$ 966                      | 9807 $\pm$ 730                    | **10388 $\pm$ 1024** |
> | Humanoid    | **10281 $\pm$ 588**                  | 10157 $\pm$ 560                   | 9521 $\pm$ 820       |
>
> Additionally, we ran the suggested experiment for TD3. The results are presented in the table below. Full learning curves are included in the PDF of the general response (Figure 3). For both tables, the results are for 5M time steps and 10 seeds. $\pm$ captures a 95\% confidence interval. While there are some performance benefits for some environments, the results do not suggest that the batched updates are the main reason for the performance benefit.
>
> |             | TD3 (checkpoints, checkpoint policy) | TD3 (checkpoints, current policy) | TD3 (no checkpoints) |
> |-------------|--------------------------------------|-----------------------------------|----------------------|
> | HalfCheetah | 14075 $\pm$ 1064                     | 13946 $\pm$ 1022                  | **14337 $\pm$ 1491** |
> | Hopper      | 3366 $\pm$ 362                       | 3285 $\pm$ 578                    | **3682 $\pm$ 83**    |
> | Walker2d    | **5189 $\pm$ 405**                   | 5126 $\pm$ 386                    | 5078 $\pm$ 343       |
> | Ant         | **6168 $\pm$ 677**                   | 6149 $\pm$ 713                    | 5589 $\pm$ 758       |
> | Humanoid    | **5785 $\pm$ 154**                   | 5534 $\pm$ 136                    | 5433 $\pm$ 245       |
>
> **Questions:**
> 1. **Including the original input:** Correct! This is what we found during our testing (and shown in Figure 2). The input to TD7’s critic includes both the embeddings $z^{sa}$ and $z^s$ (latent state action) as well as the original state and action. The input to the actor includes the embedding $z^s$ and the original state.
> 2. **Percent loss:** Percent loss refers to $\frac{\text{performance of modification} - \text{performance of TD7}}{\text{performance of TD7}} \times 100$. TD7 uses all the default choices. The default choices would be 0 in all cases. We will include these details in the next version of the paper.
> 3. **Early training with checkpoints:** Experimentally, we looked at this in Table 11 in Appendix F.  Your intuition is correct, in that the policy (and corresponding data) improves very rapidly early in learning so batching the training means the data will be more distant from the behavior of the current policy (distribution shift), and corresponds to a lower reward.
> 4. **Relevance of data:** By relevance we mean how much the data corresponds to the current policy (how much distribution shift there is). In the standard training paradigm, after every training step, a new data point is collected. However, using checkpoints means that the training gets batched, so as the policy trains, before collecting the next batch, it may select actions which are not yet contained in the dataset. We will clarify this in the next version of the paper. Empirically we did not find this problematic for 20-50 episodes (after early learning stages), but presumably if this number was set to very large values (say, 1000), then the training paradigm would resemble offline RL and would likely negatively affect performance.

---

> > ### Comment · Reviewer_YfJA · 2023-08-12
> > **Response to author rebuttal**
> >
> > I want to thank the authors for providing thorough responses to my comments (as well as other reviewer comments). All of my comments have been addressed. I maintain my score, and would advocate for accepting this paper.
> >
> > Reviewers Casd and FeJr expressed concern about the work’s novelty, though I do not share these concerns. Joint state-action representation learning is under-explored compared to just state or action representation learning, and this work provides a thorough analysis of relevant design choices missing in the existing literature.
> >
> > Reviewer sfFv noted that the proposed improvements – when considered individually – are incremental. I think this is a valid comment regarding target Q clipping (Eq. 6) and checkpointing (Sec. 5.2), but the core contribution of this work is the novel state-action representation learning method and its analysis. While target Q clipping and checkpointing can in principle be applied to many RL algorithms without SALE, it serves a particular purpose in the context of SALE which the authors mention in section 5.1. The policy and Q function are more prone to extrapolation error because SALE increases the dimensionality of their inputs, and these techniques help mitigate this effect.

---

> > > ### Author Response · Authors · 2023-08-14
> > > **Thank you!**
> > >
> > > Thank you for advocating for the paper. We agree with your comments and appreciate your support.

---

### Official Review · Reviewer_sfVv · 2023-07-11

**Soundness:** 3 good
**Presentation:** 4 excellent
**Contribution:** 3 good
**Rating:** 7
**Confidence:** 4

**Summary:**

This paper proposes TD7, an improved version of the popular TD3 algorithm with 4 additional techniques: state-action learned embeddings (SALE, the major one among the four), using policy checkpoints for stable evaluation, an existing prioritized experience replay method called LAP, an existing offline RL algorithm called TD3+BC. The four techniques strengthen vanilla TD3 algorithm from different aspects. TD7 is evaluated on MuJoCo and D4RL benchmarks for online and offline RL settings, significantly outperforming existing continuous control algorithms. Comprehensive ablation studies are also included.

**Strengths:**

- This paper is well written and the content is clear.
- The proposed improvements (almost) make sense to me and well motivated or explained.
- The part of SALE (Section 4) is interesting to me, although SALE can be viewed as an improvement over OFENet. I think the detailed studies on useful techniques for stable representation learning could be inspiring to other similar problems in RL.
- This paper conduct ablation studies for most design choices to rule out the alternative choices empirically.
- The code implementation is neat and I assume it is easy to reproduce the experiments.

**Weaknesses:**

- The proposed improvements are piecemeal, each of which is incremental or existing.
- A few additional hyperparameters (e.g., the dimensionality of $z^{s,a}$, the episode number to maintain the policy unchanged) are introduced, although the authors give recommended values.
- Although the paper has comprehensive empirical studies, intuitive explanations are lacked.

**Questions:**

TD7 needs to keep current policy fixed for several episodes:

1. What if the episode horizon is long?
2. Will this be inconvenient to combine high update frequency (like used in REDQ)?
3. How to consider the gap between using the checkpoint policy for evaluation (exploitation) while the comparison and selection of checkpoint policies are based on the performance of exploration policies?

**Limitations:**

- The proposed improvements seems to be general and orthogonal to different continuous control RL algorithms. However, SALE is not combined with SAC, TQC and REDQ to evaluate the generality.
- TD7 is evaluated on MuJoCo and D4RL whose backends are both OpenAI MuJoCo suite. I would be interested in seeing TD7 in other continuous control benchmarks like DMC.

---

> ### Author Rebuttal · Authors · 2023-08-08
>
> Thank you for your comments and considerate questions.
>
> **Intuitive explanations:** Space was obviously a bit of a concern with the paper (and fortunately the camera-ready allows for an additional page). We would be happy to expand on some of the intuition/reasoning behind our improvements with this additional space. We are happy to discuss details during the rebuttal phase if there are any specific improvements you have questions about.
>
> **Checkpoint questions:** It’s worth highlighting that checkpoints are not a fundamental aspect of TD7, as they are easy to remove. We ablate over the usage of checkpoints in Figure 5 and more extensively in Appendix F. Notably, we still outperform existing state-of-the-art methods without the use of checkpointing. In situations where checkpoints are problematic, there does exist the option to not use them. However to answer your questions:
> - **Long episode horizon:** This is a good thought. Ultimately any method which relies on any kind of MC estimate could have problems for long horizons. There are some possible modifications to checkpoints such as only considering a finite/shorter horizon when evaluating the checkpoint, or similarly, accounting for the discount factor, or simply evaluating with fewer episodes. Another possibility is to use an off-policy evaluation method rather than using raw values. Ultimately there’s a lot of room for new research for unique settings.
> - **Combining with high frequency updates:** We don’t believe this is an issue, although there is some potential for conflict if the number of updates is very significant, say 1000x, then training approaches the offline setting. For smaller increases (say 2-5x), our empirical analysis (in Appendix F) did not suggest that there were any stability or training issues with training for a large number of updates before collecting new data. In practice, since our method trains after an episode finishes (rather than during the episode), it could be argued that our method is better suited for expensive training methods. This is because training the networks during an episode could cause practical issues with latency.
> - **The gap between the checkpoint policy and the exploration policy:** Great question! We don’t really cover this in the main paper due to a lack of space, but discuss it more in Appendix F. In practice, we use the minimum performance over N episodes, rather than the mean (quantiles could also be used for particularly stochastic environments). Since our goal is stability, the idea is that by using the minimum we put a much higher preference on robust policies. Following that logic, the performance gap between the exploration policy (which includes noise) and the true deterministic policy just helps us choose more robust policies. Again, there are some other possible research questions which arise depending on the ultimate goal from using checkpoints but we leave that to future work.
>
> We agree with the reviewer that these are all interesting thoughts. We believe that checkpoints are a promising avenue for research and we hope to explore some of these ideas in future work.
>
> **Combining SALE with other algorithms:** Good point! We present results of SAC with SALE compared against vanilla SAC. SALE significantly improves the results of SAC in the majority of the environments. This change only represents the addition of SALE, and with no underlying hyperparameter changes to either SAC or SALE. Since TQC and REDQ are both based on SAC, this shows that SALE could offer performance benefits to those methods as well. Learning curves are included in the PDF of the general response. Results are for 5M time steps and 10 seeds. $\pm$ captures a 95\% confidence interval.
>
> |             | SAC + SALE          | SAC                |
> |-------------|---------------------|--------------------|
> | HalfCheetah | **17316 $\pm$ 236** | 15526 $\pm$ 697    |
> | Hopper      | 2842 $\pm$ 951      | **3167 $\pm$ 485** |
> | Walker2d    | **6887 $\pm$ 306**  | 5681 $\pm$ 329     |
> | Ant         | **9288 $\pm$ 462**  | 4615 $\pm$ 2022    |
> | Humanoid    | **9062 $\pm$ 335**  | 6555 $\pm$ 279     |
>
>
> **TD7 on other benchmarks:** We have gathered results on the DMC suite. As shown in the table below, TD7 consistently maintains its advantage over TD3 in DMC as well. In the attached PDF of the general response, we include the full learning curves for these environments. Results are for 1M time steps and 10 seeds. $\pm$ captures a 95\% confidence interval.
>
> |                         | TD7                 | TD3                |
> |-------------------------|---------------------|--------------------|
> | Ball in Cup Catch       | **982 $\pm$ 1.8**   | 981 $\pm$ 3.3      |
> | Cartpole Balance Sparse | **1000 $\pm$ 0.0**  | **1000 $\pm$ 0.0** |
> | Cartpole Swingup        | **875 $\pm$ 5.8**   | 871 $\pm$ 6.6      |
> | Cheetah Run             | **914 $\pm$ 2.9**   | 866 $\pm$ 23.5     |
> | Finger Spin             | **989 $\pm$ 1.4**   | 975 $\pm$ 14.1     |
> | Finger Turn Hard        | **742 $\pm$ 136.8** | 552 $\pm$ 80.7     |
> | Hopper Stand            | **774 $\pm$ 177.4** | 480 $\pm$ 180.8    |
> | Hopper Hop              | **175 $\pm$ 98.7**  | 58 $\pm$ 42.3      |
> | Humanoid Stand          | **504 $\pm$ 336.5** | 7 $\pm$ 0.4        |
> | Humanoid Walk           | **389 $\pm$ 157.6** | 36 $\pm$ 72.4      |
> | Walker Stand            | **991 $\pm$ 1.5**   | 985 $\pm$ 3.1      |
> | Walker Walk             | **981 $\pm$ 2.7**   | 965 $\pm$ 5.2      |
> | Walker Run              | **807 $\pm$ 62.5**  | 631 $\pm$ 74.7     |
> | Quadruped Walk          | **950 $\pm$ 4.9**   | 750 $\pm$ 168.9    |
> | Quadruped Run           | **684 $\pm$ 87.5**  | 503 $\pm$ 193.7    |
> | Mean                    | **784**             | 644                |
> | Median                  | **875**             | 750                |
> | IQM                     | **856**             | 733                |

---

### Official Review · Reviewer_Casd · 2023-07-25

**Soundness:** 3 good
**Presentation:** 2 fair
**Contribution:** 2 fair
**Rating:** 5
**Confidence:** 4

**Summary:**

This paper introduces a couple of ideas to improve the empirical performance of the TD3 algorithm on continuous-action RL problems.

The core contribution is to show that learning state and action embeddings that are designed to predict themselves in successive timestep can help achieve more reward. This is pretty nice because the representation learning process is done in conjunction with reward maximization and is shown to be useful and stable enough to enable faster learning in the same task.

A few other contributions are presented which I list below:

1- a normalization approach that scales the output of the embeddings, and which is compared against some of the existing normalization approaches such as batch and layer norm.

2- checkpointing the policy during online RL, akin to supervised learning, so rather than using the latest learned policy one can continue learning by using the best-performing policy.

3- clipping the value estimates during training to ensure that the estimates remain within a meaningful range.


Very nice empirical results are then provided, most notably on the Mujoco benchmark in online RL, where the proposed approach named TD7 is capable of beating competitive baselines such as TD3 and SAC.

**Strengths:**

The highlight of the paper is the impressive empirical results provided on the online RL experiments with the Mujoco baseline. I did my best to cross-check the performance of the baseline agents and it does seem like that TD7 is capable of beating TD3 and SAC (these are the two baselines I checked) on these benchmarks. I do see some discrepancy between SAC results reported here and those reported (for Hopper) in the original paper, but otherwise the results are consistent with published papers.

**Weaknesses:**

I did enjoy reading this paper, and I applaud authors for their successful implementation and empirical results. That said, I am not sure what the core message of the paper is. I think the contributions need to be motivated, framed, and highlighted better. For instance, is this paper primarily about the advantage of learning state and action embeddings in RL? If yes, existing work has already demonstrated that, so other than the fact that the empirical results are superior, what kind of statements can we make about this new approach to learning embeddings that we did not know previously? Is it better for transfer learning? Is it faster in terms of run time? Is it better motivated theoretically?

Just to name a few papers, with the danger of missing other related and interesting work:

- An approach very similar to the one presented here was presented earlier by Zhang et al: "Learning Action Representations for Reinforcement Learning"

- Gelada et al also propose a very similar approach in "Deepmdp: Learning continuous latent space models for representation learning", and I am not clear in what sense SALE is doing something different than them.

- Chandak et al show in "Learning Action Representations for Reinforcement Learning" that one can learn a low-dimensional representation for actions.

These are just a few examples to show that the idea of learning state and action representations is well-explored and to me this limits the novelty of the work.


Moving to the other contributions, and in terms of the normalization approach, I am not quite positive in what way this normalization is going to hedge against the issue of collapsing all states and actions to 0. To me it seems like that if the reward is not part of the process of learning the embeddings, and we use the kind of successive predictive loss in the embedding space, then trivially the 0 solution would be optimal with or without normalization.



I found the checkpointing idea creative and interesting. That said, I have a few issues with it: 1- it seems like based on Figure 7 that checkpointing is rarely effective in the online case, and as the paper states, not applicable in the offline case. 2- it seems to me that when computing the goodness of a checkpointed policy, a potential issue is that during training we add some exploration noise to the action suggested by the network, so unless we do zero exploration, we need to account for the fact that we have not exactly executed the policy and therefore some off-policy learning is needed to compute the true goodness of the checkpointed policies. Do I get this right?

The clipping idea also makes sense, and it is interesting, but it is still a limited contribution.

**Questions:**

- In what way is the new embedding-learning approach different than the previous work? What are the advantages and disadvantages, and what can we learn about designing embedding-learning algorithms in light of your experiments?

- How is the normalization hedging against the 0 mode collapse?

- Do we not need to do off-policy learning during check-pointing to account for exploration?

---

> ### Author Rebuttal · Authors · 2023-08-08
>
> Thank you for the very detailed review, and we appreciate the highlighted positives in our work! We address your key points below.
>
> **Contribution:** We will aim to tighten up the writing in the introduction and provide a stronger and clearer message for the paper. We believe our paper adds to the current landscape of representation learning in RL by answering (or adding to) the following questions:
>
> - **What design choices should I make if I want to use representation learning for low-level states?** Almost every representation learning paper in RL is applied to image-based tasks (this includes both Zhang et al., and Gelada et al. that the reviewer mentioned). This raises some important design questions when transferring ideas from an image-based task to low-level states. What should the target in the loss function be? What should the embedding be over? Is normalization needed and if yes, how should it be used? Should learning be end-to-end or not? For each of these questions we can point to several papers which take different approaches. Our paper tests each of these choices, shows what choices matter the most, and suggests the best option.
>
> - **What challenges should I be aware of when using state-action representation learning?** While state-only representation learning is fairly common, state-action representation learning is not. Interestingly enough, we find that using this combination can introduce new challenges (i.e. extrapolation error in online RL, section 5.1) that have not been discussed or adequately addressed in prior work.
>
> - **I just want the highest performing continuous control algorithm for online or offline RL.** The MuJoCo benchmark has been the cornerstone benchmark for continuous control for many years, and many important algorithms such as PPO, TD3 and SAC have been designed around it. This space is highly saturated and improvements over TD3/SAC are either incremental or come at high computational costs. TD7 suffers from neither of these problems, as it significantly outperforms existing methods (including the aforementioned expensive ones!) and comes at a much lower run time cost than the expensive methods (for run time see Appendix I).
>
> **Related work:** There is a lot of important related work in this area and we were careful to cite all three of the papers mentioned. Our method is similar to many existing methods in that they all do representation learning from a dynamics-modelling based objective. We cite at least a dozen papers which also use this same idea. This idea appears often in the literature because it is a fundamental concept relating to Bisimulation metrics. The important question is how do we best implement this idea with complexity of deep RL where there is function approximation, continuous states and actions, finite data, etc. We believe our work adds to this important story on the empirical side.
>
> Specifically, here’s how our method differs from the mentioned approaches:
>
> - **Zhang et al.** The title listed is presumably a copy error. Our best guess is “Learning Invariant Representations for Reinforcement Learning without Reconstruction” by Zhang et al., but let us know if otherwise. The primary goal of this paper is abstraction in the state space for visual tasks. In SALE we are interested in feature learning in a low-level state space, rather than abstraction in a complex visual space. Instead of trying to summarize the input, we are trying to expand the input with useful features. We also learn a representation over state and action, rather than just the state. This distinction is important– In section 5.1 we find that doing so introduces new challenges which need to be addressed.
> - **Gelada et al.** is closely related. A few obvious differences: (setting) they work in the image space with discrete actions, (conceptual)  they only learn a representation over the state, (design) they include the reward in the target, they learn the representation end-to-end as an auxiliary loss, they don’t use a target network for the encoder. In section 4.2 in our paper, we find that the nuances in design choices can have a significant impact on performance. Similar to Zhang et al, they work in a visual space and learn state-only features.
> - **Chandak et al.** is similar in that they consider actions in their representation. However, while we consider both state and action, they don’t consider the state at all. Furthermore, their approach to representation learning is very different from ours, based on a policy gradient method, rather than learned from a disentangled signal.
>
> **Normalization:** With AvgL1Norm it is impossible for the embedding to be 0. Even for a very small $\epsilon$ we have $\frac{\epsilon}{ \frac{1}{N} \sum \epsilon} \approx 1$. Prior work does something similar by normalizing only in the loss function. It turns out that normalizing the embedding is important experimentally, since we use the embeddings concatenated with the original state-action input.
>
> **Checkpoints:** Not sure which figure you are referring to (7 is a typo presumably). Regardless, the value of checkpointing is more in stability than raw performance, since instability can get averaged out. In the general response (and the attached PDF), we show how checkpoints improve the stability of RL algorithms (Figure 4).
>
> There is indeed some bias introduced by the exploration noise used during checkpoints. However, this isn’t necessarily problematic since the checkpoint is more likely to prefer policies which are robust to noise in the action space. One issue that could arise is if the random exploration noise results in some kind of positive benefit. To combat this, we actually use the minimum performance over N episodes. This choice is discussed and evaluated further in Appendix F.

---

> > ### Comment · Reviewer_Casd · 2023-08-11
> >
> > Thanks for the added experiments, it does strengthen the case for the paper.
> >
> > I am still lingering on the effect of the proposed normalization step. Can you clearly define the optimization problem being solved here, and demonstrate how the normalization step is hedging against the network collapsing to trivial solutions?
> >
> > "However, this isn’t necessarily problematic since the checkpoint is more likely to prefer policies which are robust to noise in the action space."
> >
> > I do understand that this empirically would not be a terrible idea, however, I meant that to estimate the utility of a checkpoint policy in an unbiased way, one needs to be doing off-policy policy evaluation.

---

> > > ### Author Response · Authors · 2023-08-12
> > >
> > > Thank you for the quick response!
> > >
> > > **Embeddings**: In the context of training the embeddings, what we do is very similar to BYOL [1] and SPR [2], who only apply normalization in the loss function. The main difference between normalizing the embedding and the loss function is that the final embedding used by a downstream network (the value function in our case) will be normalized. During training, the process is effectively identical.
> > >
> > > BYOL has a discussion on collapse in Section 3.2 of their paper which is relevant to our approach. The main intuition is that while there do exist trivial uninformative solutions, there is no reason for the method to actually converge to them. This is because we are optimizing a loss towards a target network which changes independently of the loss function.
> > >
> > > Formally studying this optimization process is an important problem in both RL and self-supervised learning, but definitely out of the scope of this project. Empirically we haven’t found any issues with representation collapse, and this observation is consistent with work based on BYOL. We also experiment with some settings that are closer to what SPR/BYOL propose in the design choices section of our paper (Section 4.2) and find our approach is better for our use case.
> > >
> > > **Checkpoints**: Yes, you are certainly correct that to get an unbiased estimate of the checkpoint we need to do a form of unbiased off-policy evaluation. What our method does is use a proxy measure of the quality of the checkpoint policy (the online exploration policy performance).
> > >
> > > There are a lot of potential alternate approaches that could be used (for example, delving into OPE literature). Even naively there are simple adjustments that could be made to minimize bias, such as running a potential checkpoint for X episodes with no exploration noise if it surpasses a threshold or the previous exploration policy.
> > >
> > > However, in practice we found that our proxy was accurate and does not come at any additional costs in terms of algorithmic complexity, run time, or sample-efficiency. Regardless, we think further studying the use of checkpoints of RL is an exciting research direction and this is just an initial outline for a potentially very valuable strategy for RL.
> > >
> > > **References:**
> > > - [1] Grill, Jean-Bastien, et al. "Bootstrap your own latent-a new approach to self-supervised learning." Advances in neural information processing systems. 2020.
> > > - [2] Schwarzer, Max, et al. "Data-Efficient Reinforcement Learning with Self-Predictive Representations." International Conference on Learning Representations. 2020.

---

> > > > ### Comment · Reviewer_Casd · 2023-08-17
> > > > **Normalization**
> > > >
> > > > I am still not seeing why this normalization operation avoids mode collapse, which is claimed on line 123. To the best of my knowledge, the paper does not show empirically that mode collapse is avoided. The paper rather shows that it works better in terms of reward when compared to other other normalization strategies presented in prior work. Yes, I do agree that the embeddings will not go to 0, but they can technically mode collapse to trivial, constant solutions. I also was hoping a rough theory explanation could be provided, but it seems that arguing it would be difficult at this point.
> > > >
> > > > Also, the BYOL paper is providing a different intuition about why mode collapse is not manifesting itself:
> > > >
> > > > "... the combination of (i) the addition of a predictor to the online network and (ii) the use of a slow-moving average of the online parameters as the target network encourages encoding more and more information within the online projection and avoids collapsed solutions"
> > > >
> > > > To summarize, It is warranted to claim this is an empirically superior normalization approach, but the paper has not convinced me that this normalization is hedging against mode collapse. While I appreciate that the fact that this normalization works better empirically, I think the question about why mode collapse is not happening (and how to further reduce its possibility) is an open one.

---

> > > > > ### Author Response · Authors · 2023-08-17
> > > > >
> > > > > Thank you for the response!
> > > > >
> > > > > We believe our statement is true based on empirical results and well established prior work. However, we also agree with the reviewer that it is best not to make claims that we cannot theoretically defend within the scope of the paper, so will adjust the statements on line 123 to remove any comments on representation collapse:
> > > > >
> > > > > > Similar to the normalized loss functions used by SPR [Schwarzer et al., 2020] and BYOL [Grill et al., 2020], AvgL1Norm protects from monotonic growth, but also keeps the scale of the downstream input constant without relying on updating statistics (e.g. BatchNorm [Ioffe and Szegedy, 2015]).

---

> > > > > > ### Comment · Reviewer_Casd · 2023-08-17
> > > > > > **Mode Collapse**
> > > > > >
> > > > > > Thanks for acknowledging this. I am going to raise my score from 4->5, but just to be clear, other than the absence of theoretical evidence, empirical evidence is only that this normalization step is working better than the alternatives, meaning that it is more performant (Figure 2 from the submission). As to why exactly this is the case we are not sure yet, right?
> > > > > >
> > > > > > An empirical evidence for the statement would be one that shows mode collapse is actually less frequent under the proposed normalization, which having again checked the submission and the appendix, I don't think that's provided. Hope I'm not missing anything.

---

### Author Rebuttal · Authors · 2023-08-08

We would like to thank all the reviewers for their comments, suggestions for improvement, and interest in the paper.

Overall, there were two main comments that were repeated among the reviewers which were:

**Does TD7 work on other benchmarks?** To answer this, we ran TD7 on 15 new environments from the DeepMind Control suite and benchmarked it against TD3 (**Figure 1**). The final results at 1M time steps are shown below.

|                         | TD7                 | TD3                |
|-------------------------|---------------------|--------------------|
| Ball in Cup Catch       | **982 $\pm$ 1.8**   | 981 $\pm$ 3.3      |
| Cartpole Balance Sparse | **1000 $\pm$ 0.0**  | **1000 $\pm$ 0.0** |
| Cartpole Swingup        | **875 $\pm$ 5.8**   | 871 $\pm$ 6.6      |
| Cheetah Run             | **914 $\pm$ 2.9**   | 866 $\pm$ 23.5     |
| Finger Spin             | **989 $\pm$ 1.4**   | 975 $\pm$ 14.1     |
| Finger Turn Hard        | **742 $\pm$ 136.8** | 552 $\pm$ 80.7     |
| Hopper Stand            | **774 $\pm$ 177.4** | 480 $\pm$ 180.8    |
| Hopper Hop              | **175 $\pm$ 98.7**  | 58 $\pm$ 42.3      |
| Humanoid Stand          | **504 $\pm$ 336.5** | 7 $\pm$ 0.4        |
| Humanoid Walk           | **389 $\pm$ 157.6** | 36 $\pm$ 72.4      |
| Walker Stand            | **991 $\pm$ 1.5**   | 985 $\pm$ 3.1      |
| Walker Walk             | **981 $\pm$ 2.7**   | 965 $\pm$ 5.2      |
| Walker Run              | **807 $\pm$ 62.5**  | 631 $\pm$ 74.7     |
| Quadruped Walk          | **950 $\pm$ 4.9**   | 750 $\pm$ 168.9    |
| Quadruped Run           | **684 $\pm$ 87.5**  | 503 $\pm$ 193.7    |
| Mean                    | **784**             | 644                |
| Median                  | **875**             | 750                |
| IQM                     | **856**             | 733                |

Results are for 1M time steps and 10 seeds. $\pm$ captures a 95% confidence interval. Our results conclusively demonstrate that the improvements we have proposed to TD3 allow TD7 to consistently outperform it on this new benchmark of varied tasks. Between these new results, our online MuJoCo results, and our offline D4RL results, we hope that this convinces the reviewers that TD7 (and SALE) provides a consistent and meaningful improvement over TD3 in a wide range of tasks and settings.

**Does SALE work with other RL algorithms?** To answer this, we applied SALE to SAC, making no hyperparameter changes or additional modifications to SALE or SAC (**Figure 2**). The final results at 5M time steps are shown below.

|             | SAC + SALE          | SAC                |
|-------------|---------------------|--------------------|
| HalfCheetah | **17316 $\pm$ 236** | 15526 $\pm$ 697    |
| Hopper      | 2842 $\pm$ 951      | **3167 $\pm$ 485** |
| Walker2d    | **6887 $\pm$ 306**  | 5681 $\pm$ 329     |
| Ant         | **9288 $\pm$ 462**  | 4615 $\pm$ 2022    |
| Humanoid    | **9062 $\pm$ 335**  | 6555 $\pm$ 279     |

Results are for 5M time steps and 10 seeds. $\pm$ captures a 95% confidence interval. On 4 of the 5 tasks, SAC+SALE significantly outperforms vanilla SAC. Since most off-policy RL algorithms are based on either TD3 or SAC, we hope that this demonstrates that SALE can be applied more widely than just TD3.

**Additional experiments on checkpoints:** Some of the reviewers had some additional thoughts on checkpoints that we thought could be best answered with additional visualization.

In **Figure 3** we looked to answer a question posed by reviewer YfJA, which was: how does the change to batched training affect performance? This figure shows the performance of the checkpoint policy taken from TD3 trained with checkpoints, the current policy obtained simultaneously from TD3 trained with checkpoints, and vanilla TD3. The result shows that batching training can slightly improve performance in some cases (Ant), but largely has limited impact on performance.

This raises another question (which was asked by reviewer Casd), which is: how does using checkpoints contribute? In **Figure 4** we show the performance of 5 individual trials (10 trials is too visually cluttered), with checkpoints and without checkpoints. What we see is that while the average performance of the current policy is similar to the checkpoint policy, the stability of the checkpoint policy is much higher. For those who have not examined individual trials before, the instability of the current policy of TD7 may be surprising, however this is consistent with other algorithms as well. In Appendix F, Figure 18, we plot the learning curve of a single seed of TD3.

This stability benefit is perhaps better understood numerically. In the following table we present the average standard deviation over the last 20 evaluations (corresponding to the final 100k time steps of training). In brackets we include the [min performance, max performance] over those evaluations.

|             | TD7    (checkpoints) | TD7 (no checkpoints) | TD3 (checkpoints)  | TD3 (no checkpoints) |
|-------------|----------------------|----------------------|--------------------|----------------------|
| HalfCheetah | 14, [18149, 18202]   | 227, [17700, 18671]  | 98, [13745, 14118] | 319, [13404, 14665]  |
| Hopper      | 28, [4025, 4132]     | 573, [2200, 4224]    | 121, [3109, 3571]  | 344, [2412, 3727]    |
| Walker2d    | 83, [7108, 7408]     | 607, [4315, 6685]    | 26, [5124, 5213]   | 538, [3159, 5270]    |
| Ant         | 440, [8763, 10223]   | 609, [8748, 10676]   | 196, [5568, 6280]  | 716, [3287, 5709]    |
| Humanoid    | 422, [9414, 10808]   | 1251, [5213, 10501]  | 149, [5335, 5882]  | 158, [5045, 5658]    |

What we see is that checkpoints significantly reduce the variability of performance between evaluations, and improve the worst case performance.

We hope this addresses the concerns of the reviewers and are happy to discuss further.

---

### Decision · Program_Chairs · 2023-09-21

**Decision:**

Accept (poster)

**Comment:**

The paper proposes a method for learning state-action representations for continuous control off-policy deep RL and integrates it into the TD3 algorithm to instantiate the TD7 algorithm.

Main Strengths: Reviewers appreciated novelty in the under-explored area of state-action representation learning. Representatio learning methods aside from state representation learning are novel and interesting to the RL community. The paper has strong empirical evaluation including ablation studies that detail the utility of these techniques across a variety of benchmarks and algorithm settings.

Main Concerns: The main weakness noted by the reviewers is that a variety of techniques are introduced in the method and intuition and theory for these is sometimes limited.

Overall, the study of the interplay between state and action in an agents representation is interesting and valuable to the RL community and so I recommend acceptance.